



# Last Millennium Volcanic Forcing and Climate Response using SO₂ Emissions

Lauren R. Marshall[1,2], Anja Schmidt[1,3,4], Andrew P. Schurer[5], Nathan Luke Abraham[1,6], Lucie J. Lücke[5], Rob Wilson[7], Kevin J. Anchukaitis[8], Gabriele C. Hegerl[5], Ben Johnson[9], Bette L. Otto-Bliesner[10], Esther C. Brady[10], Myriam Khodri[11], Kohei Yoshida[12]

[1]Yusuf Hamied Department of Chemistry, University of Cambridge, Cambridge, UK
[2]Department of Earth Sciences, Durham University, Durham, UK
[3]Institute of Atmospheric Physics (IPA), German Aerospace Center (DLR), Oberpfaffenhofen, Germany
[4]Meteorological Institute, Ludwig Maximilian University of Munich, Munich, Germany
[5]School of Geosciences, University of Edinburgh, Edinburgh, UK
[6]National Centre for Atmospheric Science, UK
[7]School of Earth and Environmental Sciences, University of St Andrews, St Andrews, UK
[8]School of Geography, Development and Environment and Laboratory of Tree-Ring Research, University of Arizona, Tucson, AZ, USA
[9]Met office, Exeter, UK
[10]Climate and Global Dynamics Laboratory, NSF National Center for Atmospheric Research, Boulder, CO, USA
[11]LOCEAN/IPSL, Sorbonne University/IRD/CNRS/MNHN, Paris, France
[12]Department of Atmosphere, Ocean, and Earth System Modeling, Meteorological Research Institute (MRI), Japan Meteorological Agency, Tsukuba, Japan

*Correspondence to*: Lauren Marshall (lauren.marshall@durham.ac.uk), Anja Schmidt (anja.schmidt@dlr.de)

**Abstract.** Climate variability in the last millennium (past 1000 years) is dominated by the effects of large-magnitude volcanic eruptions; however, a long-standing mismatch exists between model-simulated and tree-ring derived surface cooling. Accounting for the self-limiting effects of large sulfur dioxide (SO₂) injections and the limitations in tree-ring records such as lagged responses due to biological memory reconciles some of the discrepancy, but uncertainties remain particularly for the largest tropical eruptions. The representation of volcanic forcing in the latest generation of climate models has improved significantly, but most models prescribe the aerosol optical properties rather than using SO₂ emissions directly and including interactions between the aerosol, chemistry and dynamics. Here, we use the UK Earth System Model (UKESM) to simulate the climate of the last millennium (1250-1850) using volcanic SO₂ emissions. Averaged across all large-magnitude eruptions, we find similar Northern Hemisphere (NH) summer cooling compared with other last millennium climate simulations from the Paleo Model Intercomparison Project Phase 4, run with both SO₂ emissions and prescribed forcing, and a continued overestimation of surface cooling compared with tree-ring reconstructions. However, for the largest-magnitude tropical eruptions in 1257 (Mt. Samalas) and 1815 (Mt. Tambora), some models including UKESM1 suggest a smaller NH summer cooling that is in better agreement with tree-ring records. In UKESM1, we find that the simulated volcanic forcing differs considerably from the PMIP4 dataset used in models without interactive aerosol schemes, with marked differences in the hemispheric spread of the aerosol, resulting in lower forcing in the NH when SO₂ emissions are used. Our results suggest that for the largest tropical eruptions, the spatial distribution of aerosol can account for some of



the discrepancies between model-simulated and tree-ring derived cooling. Further work should therefore focus on better resolving the spatial distribution of aerosol forcing for past eruptions.

## 1 Introduction

Large-magnitude, stratospheric $SO_2$-injecting volcanic eruptions are the dominant cause of surface temperature variations over the last millennium prior to the rise of anthropogenic greenhouse gases (Jungclaus et al., 2017; Schurer et al., 2013; Schurer et al., 2014), but the magnitude, duration, and spatial structure of the radiative forcing from volcanic sulfate aerosol and that of the surface cooling, are subject to large uncertainties. Our understanding of the climate of the last millennium is based on proxy reconstructions and climate model simulations, yet climate model simulations have tended to overestimate

the surface cooling following volcanic eruptions compared with tree-ring reconstructions (Wilson et al., 2016; Timmreck et al., 2021; Schneider et al., 2017; Hartl-Meier et al., 2017; Masson-Delmotte et al., 2013) as well as instrumental observations (e.g., Marotzke and Forster, 2015), although internal climate variability including the El Niño Southern Oscillation (ENSO), can account for some of the differences over the instrumental period (Lehner et al., 2016; Schurer et al., 2023).

There are several possible non-exclusive explanations that could account for the last millennium mismatch including that the calculated volcanic forcing is too strong, that there are errors in dating eruptions, that the models are too sensitive to the forcing, and that the proxy reconstructions are biased. Past volcanic emissions and volcanic forcing are derived from sulfate in ice cores. The ice core records are composited to produce an average of the sulfate deposited over the ice sheet, which is then converted to estimates of volcanic sulfur dioxide ($SO_2$) emissions and further converted to aerosol optical properties,

such as the aerosol extinction, single scattering albedo and the asymmetry factor using simple scalings and/or simple transport models (e.g., Gao et al., 2008; Crowley and Unterman, 2013; Toohey and Sigl, 2017). This output is then used to prescribe the aerosol optical properties and forcing for eruptions in climate model simulations without interactive sulfur chemistry and aerosol microphysical schemes (Jungclaus et al., 2017). The volcanic forcing could be too strong if the estimated $SO_2$ emission was too large or due to uncertainties in the conversion of the emission into the optical properties,

which is largely based on the relationship observed and simulated in models following the 1991 eruption of Mt. Pinatubo (Crowley and Unterman, 2013; Gao et al., 2008; Toohey and Sigl, 2017). However, these relationships between ice sheet deposited sulfate, emissions and optical properties are unlikely to hold for all eruptions (Marshall et al., 2021) and consequently there is a large uncertainty associated with past forcing estimates. Volcanic eruptions with large injections of $SO_2$ lead to larger sulfate aerosol particles (through condensation and coagulation) that limit the magnitude of the climatic

response due to a reduced scattering efficiency and a higher aerosol removal rate (e.g., Pinto et al., 1989). Climate models with aerosol microphysical schemes that simulate an eruption using an initial emission of $SO_2$ and account for aerosol growth and removal have demonstrated this limiting effect, with a smaller simulated surface cooling for large eruptions (e.g., Timmreck, 2012). For the largest tropical eruptions during the last millennium – the 1257 Mt. Samalas and the 1815 Mt.



Tambora eruptions - Stoffel et al. (2015) found a smaller surface cooling in simulations using a 2D aerosol microphysical

model that better agreed with tree-ring reconstructions, compared with simulations run with older volcanic forcing reconstructions where aerosol growth and subsequent self-limiting effects were either not accounted for (Gao et al., 2008), or based on a simple scaling (Crowley and Unterman, 2013). This study also found a slightly stronger cooling in their tree-ring reconstruction. The most recent volcanic forcing estimate for the last millennium, EVA(eVolv2k) (Easy Volcanic Aerosol forcing generator run with the eVolv2k dataset for $SO_2$ emissions; Toohey and Sigl, 2017; Toohey et al., 2016a), which is

the recommended dataset for last millennium simulations conducted as part of the Paleo Model Intercomparison Project (PMIP4; Jungclaus et al. 2017), does account for the reduced forcing efficiency of sulfate aerosol for larger $SO_2$ emissions also by applying a scaling (an idealized 2/3 power law for eruptions greater than 1815 Mt. Tambora (~60 Tg $SO_2$)) and has updated eruption dates and improved spatial coverage. Although it provides a consistent reconstruction to be used across models without aerosol microphysics schemes, this forcing reconstruction does not explicitly include many chemical (e.g.,

the depletion of hydroxyl radicals), or dynamical (e.g., aerosol lofting and dispersion) and microphysical processes (e.g., the growth and removal of a population of aerosols that is not reflected in a single power law relationship) that may change the spatial aerosol distribution and the volcanic forcing.

Most eruptions in the last millennium are not attributed to specific volcanoes (of 127 identified eruptions between 800 and

1890 CE, only 30 have been confidently assigned to the volcano of origin) and hence the eruption location, eruption season and emission altitude are unknown. Assumptions must therefore be made (see Sect. 2.1.1) when creating the reconstructions for assigning the date and eruption source parameters, which affects the magnitude and spatial pattern of the volcanic forcing (e.g., Marshall et al., 2019; Toohey et al., 2011; Toohey et al., 2019). By accounting for some of the uncertainties in dating as well as in the magnitude of the $SO_2$ emissions, Lücke et al. (2023) using a simple response model demonstrated better

model-tree-ring matches for some eruptions. It is also possible that the volcanic forcing could be overestimated for some eruptions due to our lack of knowledge of co-emissions of other volcanic gases such as halogens and water vapour that can affect the forcing (Staunton-Sykes et al., 2021; Legrande et al., 2016). In particular, the warming effect of water vapour has recently been demonstrated following the 2022 eruption of Hunga Tonga-Hunga Ha'apa (Sellitto et al., 2022) such that the volcanic forcing may be incorrect for past eruptions even if the $SO_2$ emission is correct. Lastly, even if the volcanic forcing

used in climate models was correct, models may still overestimate the climate response depending on the sensitivity of the model due to its radiation scheme and climate feedbacks (e.g., Chylek et al., 2020).

There are further uncertainties associated with the tree-ring proxy records themselves (Anchukaitis et al., 2012; Büntgen et al., 2021; D'arrigo et al., 2013; Mann et al., 2012; Anchukaitis and Smerdon, 2022). In particular, differences in the proxy

response are found depending on whether the tree-ring reconstruction is based on tree-ring width (TRW) data, maximum latewood density (MXD) measurements, or both, with MXD showing a stronger cooling that aligns more closely with model simulations (Frank et al., 2007; D'arrigo et al., 2013; Esper et al., 2015; Anchukaitis and Smerdon, 2022). Zhu et al. (2020)



demonstrated that the discrepancy between model-simulated surface cooling and tree-ring records could be partly resolved when accounting for factors related to the proxy reconstructions, such as the spatial coverage (e.g., Anchukaitis et al., 2012;

Guillet et al., 2017; Büntgen et al., 2022), biological memory in TRW vs. MXD (e.g., Esper et al., 2015; Lücke et al., 2019), the season that the proxy reconstructions represent (Anchukaitis et al., 2012), and proxy noise (Neukom et al., 2018). However, differences were still found for the largest tropical eruptions in the last millennium (1257 Mt. Samalas and 1815 Mt. Tambora).

Several climate models are now simulating volcanic eruptions using complex 3D aerosol and sulfur chemistry schemes (e.g., Timmreck et al., 2018; Quaglia et al., 2023), but prior to CMIP6 no models had conducted long transient simulations of the last millennium using this approach. Studies have instead focused on the satellite era and have demonstrated good skill compared to observations (e.g., Mills et al., 2017; Mills et al., 2016; Schmidt et al., 2018; Dhomse et al., 2020). In this study, we use version 1 of the UK Earth System Model (UKESM1) to run three ensemble member simulations of most of the last

millennium (between 1250-1850) using $SO_2$ emissions directly to represent volcanic eruptions. Our simulations provide a substantial improvement compared to earlier models for realistically simulating volcanic eruptions over the last millennium. They are, however, computationally expensive, with each simulation taking over one physical year to run. Here we focus on the simulated NH (40-75°N) summer (May-August; MJJA) land surface cooling and compare our UKESM1 output with tree-ring reconstructions and with other models that have simulated the same period. Our study aims to answer the following

questions:

1. How does the simulated volcanic forcing in models with interactive aerosol capabilities compare with the EVA(eVolv2k) dataset, which is widely used in PMIP4 models without these capabilities?
2. How does the simulated NH summer cooling in PMIP4 models that prescribe the aerosol properties using
EVA(eVolv2k) compare to those simulating the aerosol interactively using $SO_2$ emissions?
3. Does using interactive aerosol modelling with the latest generation of complex aerosol climate models help to reconcile discrepancies between simulated and reconstructed cooling for the largest tropical eruptions?
4. Using the same model framework (UKESM1), how does simulated NH summer cooling differ between the two methods of volcanic forcing implementation ($SO_2$ emissions vs. prescribing the optical properties from
EVA(eVolv2k)) for the largest tropical eruptions?

## 2 Methods

### 2.1 Climate Models

An overview of the climate models used in this study is presented in Table 1. Five models that ran the PMIP4 last millennium simulation were available: MRI-ESM2, MIROC-ES2L, MPI-ESM1-2-LR, IPSL-CM6A-LR and



CESM2(WACCM6ma), all of which are complex Earth System Models. Of these, two also used $SO_2$ emissions (CESM2(WACCM6ma) and MRI-ESM2), and the remaining (MIROC-ES2L, MPI-ESM1-2-LR and IPSL-CM6A-LR) prescribed the volcanic aerosol optical properties using EVA(eVolv2k). The $SO_2$ dataset used in both UKESM1 and CESM2(WACCM6ma) is described below. Further detail on how the $SO_2$ emissions were implemented in MRI-ESM2 is included in Supplementary Text 1. To minimise computational cost, our UKESM1 simulations start in 1250, and we

therefore focus on the period 1250-1849 across all models (standard last millennium PMIP4 simulations start in 850). Our analysis focusses on the surface air temperature and stratospheric aerosol optical depth. Model data are analysed on their native grids.

**Table 1. Climate models analysed in this study.**

| Model | Volcanic forcing | No. of ensemble members | Reference |
|---|---|---|---|
| MRI-ESM2 | $SO_2$ injections from eVolv2k. See Supplementary Text 1 for further detail. | 1 | Yukimoto et al. (2019) |
| MIROC-ES2L | Aerosol optical properties from EVA(evolv2k) | 1 | Ohgaito et al. (2021) |
| MPI-ESM1-2-LR | Aerosol optical properties from EVA(eVolv2k) | 2 | Van Dijk et al. (2022); Mauritsen et al. (2019) |
| IPSL-CM6A-LR | Aerosol optical properties from EVA(eVolv2k) | 1 | Lurton et al. (2020); Boucher et al. (2020) |
| CESM2(WACCM6ma) | $SO_2$ injections from eVolv2k plus temporal emissions for 1783 Laki (see Figure S4) | 1 | This study. Danabasoglu et al. (2020) |
| UKESM1 | $SO_2$ injections from eVolv2k plus temporal emissions for 1783 Laki (see Figure S4 and Supplementary Text 2) | 3 | This study. Sellar et al. (2019) |


### 2.1.1 UKESM1

We used version 1 of the UK Earth System Model (UKESM1), a state-of-the-art model that includes an atmosphere-land-ocean-sea ice model, terrestrial and ocean biogeochemistry, and a comprehensive aerosol and chemistry scheme (Archibald et al., 2020; Mulcahy et al., 2020; Sellar et al., 2019). Two modifications from the original UKESM CMIP6 preindustrial

release job were made: the first was to simulate stratospheric volcanic eruptions interactively using the GloMAP-mode aerosol microphysical scheme (Mann et al., 2010), removing the average background aerosol forcing file. The second was to correct a bug in the aerosol scheme that changes the sulfuric acid vapour as applied in version 1.1 of UKESM (Mulcahy et



al., 2023; Ranjithkumar et al., 2021). The GloMAP-mode aerosol microphysical scheme has been used in many studies of volcanic eruptions in various configurations (Wade et al., 2020; Dhomse et al., 2014; Dhomse et al., 2020; Marshall et al.,

2021). Here we used the version of GloMAP as included in the release version of UKESM1, which is the same as that applied in Aubry et al. (2021), Visioni et al. (2022) and Chim et al. (2023) but is slightly different to that used in Dhomse et al. (2020) in which three large volcanic eruptions were recently evaluated. The main differences include a dependency of the $H_2SO_4$ condensation on the vapour pressure deficit (Dhomse et al., 2014) and the evaporation of sulfate aerosol at high altitudes that is not included in the release version of UKESM1, nor do we simulate sulfuric-meteoric smoke particles

(Brooke et al., 2017; Marshall et al., 2018). In our version, mode-merging between the accumulation and course soluble models is also deactivated above 100 hPa rather than being turned off entirely (e.g., Dhomse et al., 2014; Marshall et al., 2019). Overall, the aerosol schemes remain similar and simulations of 1991 Mt. Pinatubo using our setup show a very similar global mean stratospheric aerosol optical depth to Dhomse et al. (2020) (this study also used an older version of the climate model and a slightly lower injection altitude), which also compares well to observations (Figure S1).


**Transient simulations**

For simulating the last millennium, forcing files were made following the PMIP4 protocol (Jungclaus et al., 2017). Our simulations are aligned with the PMIP4 last millennium experiment but deviate slightly due to computational restrictions.

Instead of 850, our simulations start in 1250. We ran a 50-year spin-up from 1200 with the atmosphere and ocean fields initialized from the CMIP6 preindustrial control (1850 conditions). Because the sudden removal of agriculture will create empty land which the dynamic vegetation takes time to fill, it is better to start from initial conditions with less agriculture (so non-agricultural land is filled rather than suddenly created). Thus, the vegetation fractions for this spin-up were initialized from an 80 year-long UKESM spin up run with constant forcing from the year 850 run using a slightly different model

version with no interactive atmospheric chemistry (UKESM-CN). During these initial 50 years (1200-1250) the transient forcing (including emissions from the volcanic eruptions over this period) is applied. Further technical details and how the forcing was implemented in UKESM1 are included in Supplementary Text 2. We ran three ensemble members, starting from the years 1249, 1250 and 1251 from the spin up simulation. We also ran three historical simulations initialized from the end of each of these ensemble members. Output of basic climate states over the spin up and the transient simulations (including

surface temperature, top-of-the-atmosphere net downward radiative flux, precipitation, sea-ice, sea surface temperature and the Atlantic meridional overturning circulation) are included in the SI (Figures S2-S3). The global mean temperature is slightly higher compared to the UKESM1 CMIP6 preindustrial control simulation in line with the increase seen in UKESM1.1 (Mulcahy et al., 2023), likely a result of the update to the aerosol scheme and the change in how volcanic eruptions are simulated.




For the volcanic forcing, we use the SO$_2$ emissions and eruption latitudes provided in v2 of the eVolv2k dataset (Toohey and Sigl, 2017). For eruptions that are unattributed to specific volcanoes, the longitude of the emission was set to 140°E (0°E is used in MRI-ESM2). In eVolv2k the unattributed eruptions are assigned an eruption month of January. Because most eruptions are unattributed in the last millennium, this could introduce a potential bias when analysing the average response to

volcanic eruptions (c.f. Stevenson et al., 2017). We therefore chose to randomise the eruption month, assigning a month of either January, April, July or October to those eruptions (Table S2). In MRI-ESM2 unknown eruptions are simulated on the 1$^{st}$ of January following the eVolv2k dataset. For the 1783-1784 eruption of Laki, we also deviate from the eVolv2k dataset in the UKESM1 and CESM2(WACCM6ma) simulations by including daily emissions that reflect the different eruption phases with vent emissions injected between 0 and 2 km and explosive emissions between 9 and 13 km (Schmidt et al.,

2010). For all other eruptions the SO$_2$ emissions were injected between 18 and 20 km. Figure S4 and Table S2 show the resulting dataset used by both UKESM1 and CESM2(WACCM6ma). Further detail is included in Supplementary Text 2.

**Case study simulations**

Further case study simulations of the three largest eruptions (1257 Mt. Samalas, 1458 Unidentified and 1815 Mt. Tambora) were also conducted using UKESM1 in which the model is used in its preindustrial configuration (background conditions representative of 1850) but with the above changes to the aerosol scheme. For each eruption we ran simulations using the SO$_2$ emissions from Toohey and Sigl (2017) and simulations with aerosol optical properties prescribed from the EVA(eVolv2k) idealised forcing generator (Toohey et al., 2016a). EVA(eVolv2k) includes zonal mean monthly mean

datasets of the aerosol scattering, absorption and asymmetry parameter as a function of wavelength through the solar and terrestrial spectrum, which were interpolated onto the UKESM1 grid and averaged across the spectral bands of the radiation scheme. Running with both direct SO$_2$ emissions and prescribed aerosol optical properties enable a direct comparison between these two common approaches within the same modelling framework. Because the EVA(eVolv2k) dataset includes all eruptions, additional eruptions in 1260 (2 Tg SO$_2$), 1463 (1 Tg SO$_2$), 1821 (1 Tg SO$_2$), and 1822 (4 Tg SO$_2$) are included

in the UKESM1 prescribed simulations but not the emissions-driven. However, these additional eruptions are much smaller in magnitude and occur following the majority, if not all, of the volcanic forcing from the first eruption and consequently do not affect our main comparisons. For each eruption scenario we ran nine ensemble members initialized during different ENSO and Quasi Biennial Oscillation (QBO) phases chosen from the UKESM1 CMIP6 preindustrial control simulation (both ensembles start from the same conditions). We also ran nine equivalent control simulations with no stratospheric

volcanic SO$_2$ emission for the interactive runs, and with average historical SAOD (1850-2014) (as used in the UKESM preindustrial control simulation) in the prescribed runs because the EVA(eVolv2k) forcing dataset includes an additional background SAOD. Anomalies are calculated relative to the average of the 9 control simulations for each ensemble. The background SAOD values in EVA(eVolv2k) (~0.003) and the historical climatology (~0.01) are not identical, but as the volcanic perturbations to SAOD are orders of magnitude higher than the background, any differences due to the different





backgrounds are negligible. All runs also include a very small background (~0.001) due to aerosols from non-volcanic sources that the interactive aerosol scheme simulates in the stratosphere. The emissions-driven runs also have interactive aerosol surface area density, but climatological values are used in the prescribed simulations. The simulations were run for ten years.

## 2.2 Tree-ring reconstructions

Our analysis focusses on four recent NH tree-ring reconstructions that represent the state-of-art with respect to understanding past summer temperatures of the Common Era: NTREND2015 (Wilson et al., 2016), Büntgen2021 (Büntgen et al., 2021), Schneider2015 (Schneider et al., 2015) and Guillet2017 (Guillet et al., 2017). All records are high-resolution growing season temperature reconstructions but use different proxy networks and reflect different methodological approaches. NTREND2015 uses a network of 54 published tree-ring reconstructions across the Northern Hemisphere, 18 of which are
based entirely on maximum latewood density (MXD) and 11 use only ring-width (TRW) measurements. A further 25 records are composed of some combination of TRW, MXD and latewood Blue Intensity (BI – a proxy of latewood relative density). The NTREND reconstruction uses a temporally iterative 'composite plus scale' method and extends back to 750 CE. It is calibrated to MJJA mean land-only data from CRUTEM, averaged over the latitudes 40°N to 75°N. In contrast, Büntgen2021 uses a small network of nine long TRW with an array of methodologies to yield a 15-member ensemble of
June-August (JJA) mean temperatures that extends back throughout the Common Era. Here, we use the median of the ensemble. The Schneider2015 record utilised 15 MXD chronologies from around the NH extra-tropics and again is calibrated to JJA temperatures. MXD expresses much lower autocorrelative persistence structure than TRW and has been shown to express a much clearer and less smeared response to past volcanic forcing (Anchukaitis et al. 2012). Guillet et al. (2017) is an update of Stoffel et al. (2015) and uses a set of 25 tree-ring width and density chronologies that extend back to
the 13th century in a principal components regression approach to reconstructing June through August Northern Hemisphere (40°–90° N over land) temperature anomalies. Guillet et al. (2017) also include ice core oxygen isotope records from Greenland in their dataset. As in NTREND2015, Schneider2015 and Guillet2017 both contain wood density data and are therefore expected to express a stronger interannual temperature signal than the TRW-based Büntgen2021, but all four records have identified the signal of post-volcanic eruption cooling in their reconstructions. Because of the use of TRW data
in NTREND2015, Büntgen2021, and Guillet2017, however, the temperature recovery following the peak cooling can last up to a decade, especially in NTREND2015, whereas MXD-only reconstructions (e.g., Schneider et al. 2015) reflect a shorter period of post-eruption recovery (Anchukaitis and Smerdon 2022).



# 3 Results

## 3.1 Stratospheric aerosol optical depth 1250-1849

**Figure 1: 1250-1849 global mean monthly mean stratospheric aerosol optical depth (SAOD) at 550 nm simulated by UKESM1 (blues), CESM2(WACCM6ma) (grey) and MRI-ESM2 (orange) and from the PMIP4 dataset, EVA(2k) (black). The timeseries is split into 100-year chunks. Eruptions are marked by the vertical dashed lines and triangles. The colour of each triangle shows the**




**eruption latitude, and the size indicates the magnitude of the SO₂ emission, ranging from 0.4 Tg (for the 1512 unidentified eruption) to 118 Tg (for the 1257 eruption of Samalas). The NH average SAOD is shown in Figure S5.**

The global monthly mean SAOD timeseries from the EVA(eVolv2k) dataset (hereafter abbreviated to EVA(2k)) and as simulated by UKESM1, CESM2(WACCM6ma) and MRI-ESM2 are shown in Figure 1 and NH averages are shown in Figure S5. Zonal means are shown in Figures S6-9. There are several overall differences, although subtleties exist for individual eruptions. In general:


1. Peak global mean SAOD is similar between UKESM1 and CESM2(WACCM6ma), but higher than EVA(2k) (compare blue, grey and black lines)

2. Global mean SAOD in MRI-ESM2 (orange) (both peak and duration) is similar to EVA(2k) for most eruptions, except for 1257 Mt. Samalas, 1458 Unidentified and 1783 Laki, differing also to UKESM1 and CESM2(WACCM6ma) for 265 these eruptions.

3. Global mean SAOD in UKESM1 decays more quickly compared to EVA(2k) for the largest tropical eruptions (see for example, 1257 Mt. Samalas, 1458 Unidentified, 1815 Mt. Tambora, 1600 Huaynaputina, 1640 Parker and 1345 Unidentified, which all have SO₂ emissions greater than 30 Tg SO₂). In contrast, CESM2(WACCM6ma) simulates a much longer decay in SAOD than both UKESM1 and EVA(2k) for the larger eruptions. MRI-ESM2, although similar to 270 EVA(2k) for most eruptions, simulates a longer-lived SAOD signal than UKESM1 for Samalas, but not as long as CESM2(WACCM6ma).

4. For small-magnitude eruptions (< ~10 Tg of SO₂) UKESM1 and CESM2(WACCM6ma) simulate similar global mean SAOD in both peak values and decay timescales, which is also closer to EVA(2k).

5. Hemispheric asymmetry is different between the models and EVA(2k) for the tropical eruptions. UKESM1 simulates a 275 much stronger and confined tropical peak whereas the SAOD is more spread in CESM2 and EVA(2k) (Figures S7-10). In MRI-ESM2 there is much greater SAOD in the SH polar regions than any other model.

6. Zonal mean SAOD is more similar between UKESM1 and CESM2(WACCM6ma) for the extratropical eruptions, with higher peak values compared to EVA(2k) and MRI-ESM2.

7. Global mean SAOD in volcanically-quiescent years (i.e., background SAOD) is slightly smaller in UKESM1 and MRI- 280 ESM2 (~0.001) compared to CESM2(WACCM6ma) and EVA(2k) (~0.003).

CESM2(WACCM6ma) simulates a much higher NH SAOD than UKESM1 for 1257 Mt. Samalas, 1458 Unidentified, 1600 Huaynaputina, and 1815 Mt. Tambora despite similar peak values in the global mean, which results from differences in the spatial distribution of the sulfate aerosol following these tropical and large-magnitude (> 10 Tg SO₂) eruptions (see Figure 285 S5). The three UKESM1 ensemble members also display greater spread in the NH for some tropical eruptions because of differences in the hemispheric dispersion of aerosol due to internal variability; see for example 1286 Unidentified, 1345 Unidentified, and 1640 Parker (Figure S5). Some differences in the SAOD are also a result of the different season of





eruption for unidentified eruptions in the EVA(2k) dataset and MRI-ESM2 vs. UKESM1 and CESM2(WACCM6ma) (Sect. 2.1.1), which leads to offsets in the SAOD. See for example: 1329, 1453, 1458, 1654, 1693, 1809 and 1831. All eruptions
that are simulated in different seasons between the models are highlighted in grey in Table S2.

The 1783-1784 eruption of Laki is the only eruption where the SAOD in EVA(2k) and MRI-ESM2 is higher and longer lived than that simulated by UKESM1 and CESM2(WACCM6ma). In EVA(2k) and MRI-ESM2 the eruption is represented by a single emission of 42 Tg $SO_2$ in June 1783. In UKESM1 and CESM2(WACCM6ma), Laki is represented by several
smaller injections into the stratosphere as well as tropospheric emissions spanning 1783-1784 (see inset on Figure S4).

## 3.2 Surface air temperature 1250-1849

**Figure 2: 1250-1849 global annual mean surface air temperature (a) and NH (40-75°N) summer (MJJA) land surface air temperature (b) anomalies relative to the 1250-1849 average for each model. (c) Four NH summer tree-ring reconstructions,**
**NTREND2015 (Wilson et al., 2016), Büntgen2021 (Büntgen et al. 2021), Schneider2015 (Schneider et al., 2015) and Guillet2017 (Guillet et al., 2017). Eruptions marked as in Figure 1 (noting that given dating uncertainties, these may not be the eruption years reflected in the tree-ring reconstructions; see text).**





The differences in SAOD simulated among the three models using $SO_2$ emissions and that in the EVA(2k) dataset will result in differences in the simulated radiative forcing following the eruptions, and consequently the temperature response. Here we explore the response in UKESM1, CESM2(WACCM6ma) and MRI-ESM2 and from the three other models that have conducted last millennium simulations using EVA(2k) for the volcanic forcing (Sect. 2.1). We examine the response across the transient timeseries, in superposed epoch analyses of the average response to eruptions and that following the three largest-tropical eruptions compared to tree-ring reconstructions.

**Transient temperature**

Global annual mean and NH summer land surface air temperature anomalies are shown in Fig. 2. All models show distinct cooling after large-magnitude eruptions, with CESM2(WACCM6ma) having the strongest anomalies with a peak global annual mean cooling of -2.4°K and peak NH summer cooling of -4.0°K following 1257 Mt. Samalas. CESM2(WACCM6ma) also shows a warming between 1650 and 1750 relative to the long-term average that is not present in the other models. In the UKESM1 ensemble mean, the three strongest anomalies in NH summer land temperature are following the 1783-84 eruption of Laki, 1640 eruption and 1286 eruption, none of which have the largest $SO_2$ emissions (Table S2). The NH summer temperature from the four tree-ring reconstructions (panel 3) show that the amplitude of cooling in the proxy records, compared to the models, is substantially less, although a cooling can be found in response to most major eruptions, especially for tropical eruptions. Consistent with prior analyses, reconstructions show greater disagreements prior to 1400 (Esper et al. 2018). Guillet2017, for instance, shows a period of warming in the late 13[th] century and maintains more year-to-year variance compared to other reconstructions.

**Superposed epoch analysis**

As is common practice in many studies seeking to understand the role of volcanic eruptions on climate by increasing the signal to noise ratio, we conduct a superposed epoch analysis (SEA) (e.g., Fischer et al., 2007; Wilson et al., 2016; Schneider et al., 2017; Zhu et al., 2020; Zuo et al., 2023) to examine the average NH summer surface cooling response to the large magnitude volcanic eruptions (defined here as having $SO_2$ emissions greater than 10 Tg) in each model and in the four tree-ring reconstructions (Figure 3). To composite the different eruptions, anomalies are calculated with respect to the five years prior to each eruption and we take year 0 as the year of the eruption (other studies have also considered the year of peak aerosol load or year of peak forcing, e.g., Liu et al. 2022). For the resulting 17 eruptions (panel a), we find that all models except CESM2(WACCM6ma) simulate a similar NH summer cooling of around –0.8°K in the 1[st] year following the eruption. This consistency between the models suggests that on average, it does not matter how the eruptions are simulated in each model as the average response is similar between the prescribed and $SO_2$ emissions approaches, although there may be compensating effects due to other model differences. However, there are still differences on closer inspection. For



example, IPSL-CM6A-LR has the smallest peak NH summer anomalies and the length of the recovery from the cooling varies between the models. The average simulated peak cooling also remains stronger than the tree-ring reconstructions suggesting that a discrepancy still exists between model-simulated and reconstructed cooling despite the updated aerosol emissions/forcing dataset (Toohey and Sigl, 2017). The NTREND2015 reconstruction (solid line) has a stronger average cooling signal than the Büntgen2021 reconstruction (dashed line) but both express a prolonged recovery response. Guillet2017 (dashed-dot line), while also expressing a long recovery, has the strongest cooling response. It is likely that the longer recovery time for these three records reflects the inclusion of TRW chronologies in these composites (e.g. Esper et al., 2015; Zhu et al., 2020; Anchukaitis and Smerdon 2022), whereas Schneider2015 (dotted line) derived exclusively from MXD data shows a marginally stronger T+1 cooling than Büntgen2021 but a much quicker recovery period.

This epoch analysis, however, obscures several subtleties. It is highly dependent on the combination of eruption dates included (c.f. Rao et al. 2019, Zhu et al. 2022) as well as the anomaly reference period (i.e., anomalies in the SEA are different to the long timeseries presented in Fig. 2), and although the eruption dates are known exactly in the model simulations, given dating uncertainties, the timing of the real eruptions and their climatic consequences that were experienced by the trees may be different. This means that the models may have larger average peak anomalies due to the precisely 'known' eruption years, whereas for the reconstructions even small offsets in eruption date and timing of maximum negative forcing will smooth the response in the average. The timing of the peak cooling is also further dependent on whether the eruption was tropical or extratropical (the latter more likely leading to peak cooling in the year of eruption rather than the first year after) and the eruption season. Consequently, panels b-c, show further subsets of eruptions focussing on the tropical eruptions only. For all large-magnitude tropical eruptions (which excludes just two eruptions; 1477 Bardarbunga and 1783 Laki) there is a slight shift towards a closer match between the models and reconstructions (panel b). A further complication is that of these large events, 5 were simulated in different seasons in the UKESM1 and CESM2(WACCM6ma) simulations compared with the eVolv2k dataset (Sect. 2.1.1). Panel (c) shows only the eruptions simulated in the same season, reflecting the fairest comparison between the models. For these eruptions (which excludes 1276, 1453, 1458, 1695, 1809) the models and reconstructions appear closer still, with IPSL-CM6A-LR comparing very well with the stronger Guillet2017 record. There is still a spread among the other models and among the individual ensemble members (see blue UKESM1 lines).



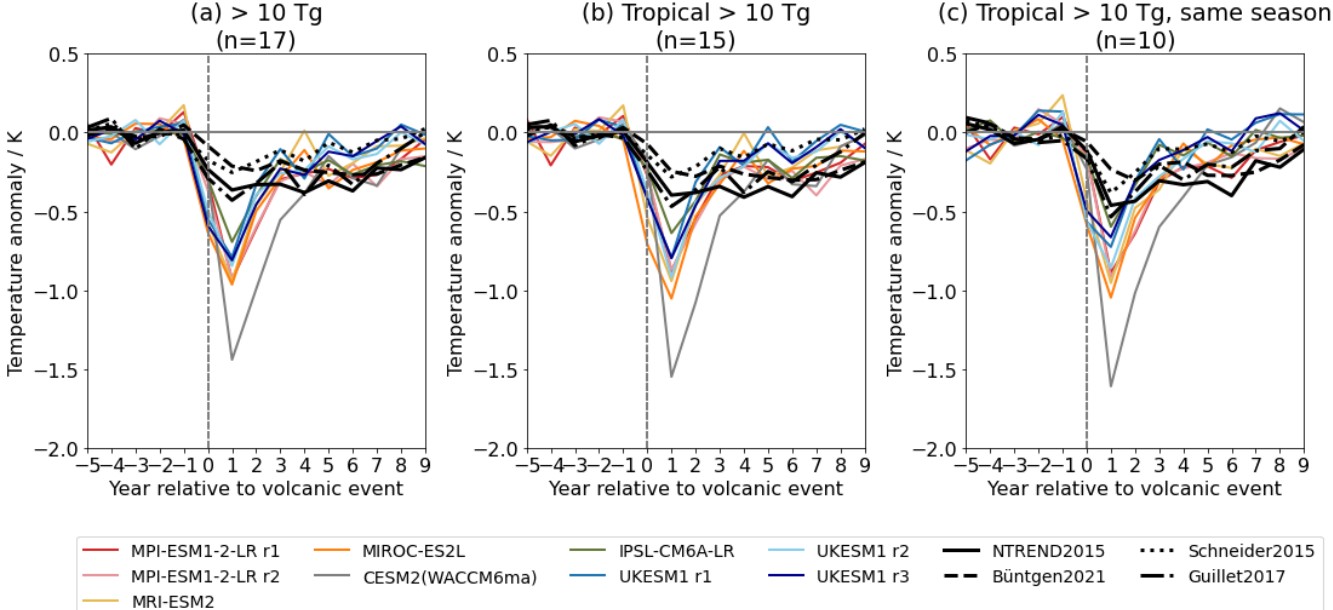

**Figure 3: Superposed epoch analysis (SEA) of NH summer surface temperature for large-magnitude eruptions (> 10 Tg SO₂). Temperature anomalies for each eruption are taken with respect to the average across the 5 years preceding the eruption, and the resulting timeseries are averaged across all eruptions. 17 eruptions are included in (a) which are: 1257, 1276, 1286, 1345, 1453, 1458, 1477, 1585, 1595, 1600, 1640, 1695, 1783, 1809, 1815, 1831, and 1835. In (b) only the tropical eruptions are included which removes 1477 and 1783. In (c) only eruptions simulated in the same season are included which removes 1276, 1453, 1458, 1695, 1809.**

Furthermore, several eruptions are closely spaced ('double events'; Toohey et al., 2016b) in which the anomalies for each eruption are affected by the other (although this should also be the case in the tree-ring reconstructions, unless there are differences in dates) (but see also Tejedor et al., 2021). This is especially important for the 1815 eruption of Mt. Tambora which follows the 1809 unidentified eruption because UKESM1 simulates a much stronger surface cooling following the 1809 eruption than Tambora (Fig. 4). Although the surface temperature following Tambora in UKESM1 is cooler than the long-term 1250-1849 average (Fig. 2, Fig. 4), anomalies with respect to the five preceding years show a warming (Fig. S10), which means that in the epoch analysis, Tambora will offset some of the average cooling for UKESM1. Figure S10 also shows anomalies for the largest three eruptions (1257 Mt. Samalas, 1458 Unidentified and 1815 Mt. Tambora) with respect to the closest 5-years unaffected by a previous eruption, demonstrating the sensitivity to the reference period for both the model and tree-ring anomalies.

For the selected eruptions shown in Fig. 3, the tree-ring anomalies do not always show a traditional peak cooling response in the first post eruption year as indicated by the models, and also have a longer smoother tail (see above). To demonstrate how date uncertainty could influence these anomalies, Fig. S11 shows an epoch analysis for eruption dates optimised to the cooling signals seen in the tree-ring reconstructions, which results in a stronger first year post eruption cooling (but does not





remove the prolonged recovery). Lücke et al. (2023) also demonstrated the importance of eruption dating uncertainty, which can make a significant difference to the amplitude of the average response shown in the SEA.

**Temperature response following the three largest tropical eruptions**

The SEA also masks differences between the models for individual events and with the comparison to tree-ring records. There are several eruptions that stand out in which the volcanic cooling (both peak and duration) is different among the models (Fig. 2). Examples include the three largest eruptions, which we subsequently focus on in the remainder of this study,

but also other eruptions in 1510 (see MIROC-ES2L and the two MPI-ESM1-2-LR ensemble members), 1600 (Huaynaputina; see MIROC-ES2L and IPSL-CM6A-LR) and 1783-84 (Laki; see all models). These differences demonstrate not only differences between volcanic forcing implementation, but also between the different models and their ensemble members.

The global annual mean and NH summer anomalies for 1257 Samalas, the 1450's eruptions and the 1809 and 1815 eruptions are shown in Fig. 4. In the global mean, IPSL-CM6A-LR (green; prescribed forcing) has the smallest initial cooling after 1257 Samalas. UKESM1 (blues; $SO_2$ emissions) simulates slightly less cooling for 1257 than the remaining models, which recovers more quickly, also for 1458 (Fig. 4a and b). In general, this is also the case for 1815, except the anomalies are comparable to the cooling from MRI-ESM2 (yellow; $SO_2$ emissions). For NH summer cooling following Samalas, both

UKESM1 and IPSL-CM6A-LR simulate cooling that compares well with the range of tree-ring responses (less than -1°C) compared with the other models that simulate peak cooling in excess of -1.8°C. If, however, the anomalies are calculated with respect to the preceding five years (Fig. S10) the simulated cooling in IPSL-CM6A-LR is slightly stronger than UKESM1. The peak cooling in the Guillet2017 and Büntgen2021 records also appears in the second year following the eruption as is also the case in the first MPI-ESM1-2-LR ensemble member. In the tree-ring reconstructions, this difference

likely reflects the inclusion and location of specific proxy datasets relative to the spatiotemporal pattern of climate anomalies (Anchukaitis et al. 2012, Anchukaitis et al. 2017) as well as the implicit weighting given to the proxy data by the reconstruction methods.





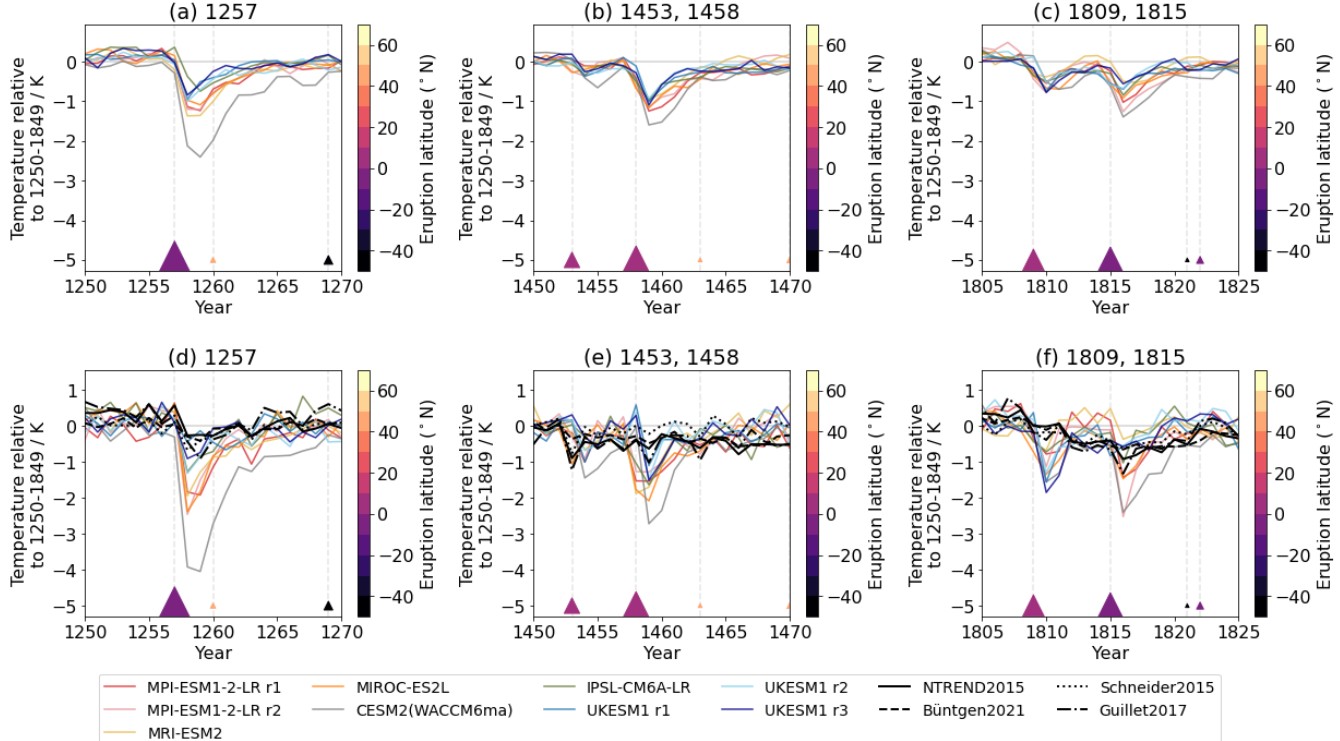

**Figure 4: Global annual mean surface air temperature (a-c) and NH summer land surface air temperature (d-f) anomalies (as in Figure 2; relative to 1250-1849 average) for the three largest eruptions as simulated by the six models and in the tree-ring reconstructions.**

For the 1458 eruption, peak NH summer anomalies in UKESM1 are similar to the other models, with ensemble member 1 having the smallest peak cooling. There is a temporal offset in the anomalies between UKESM1 and CESM2(WACCM6ma) and the other models as this eruption was simulated in July in UKESM1 and CESM2(WACCM6ma), rather than in January (the 1453 eruption is also simulated in October, rather than January). Except for Guillet2017, which includes ice core oxygen isotope data from Greenland, the reconstructions for this period also show a sustained cooling following the 1453 eruption, but not an additional cooling spike following 1458 as suggested by the models (c.f. Esper et al., 2017).

All models show a strong immediate NH summer cooling following the 1809 eruption, which is not reflected in the hemispheric-mean tree-ring reconstructions, also reported by Timmreck et al. (2021) for MPI-ESM1-2-LR, and which could be due to an overestimated $SO_2$ emission for this eruption. The identity, location and exact timing of this eruption also remain unknown, which could further account for some of the differences between model simulations and data. Tree-ring reconstructions record spatially and temporally variable temperature responses following the eruption (e.g., Anchukaitis et al., 2017; King et al., 2021; Leland et al., 2023), perhaps as a result of internal ocean-atmosphere variability, which could dampen the cooling signal in the large-scale mean. For all models except UKESM1 and MRI-ESM2, there is a second



stronger cooling following 1815 Mt. Tambora. UKESM1 indicates a sustained cooling over this period, but not an additional cooling anomaly following Tambora. The 1809 eruption was also simulated in October in UKESM1 and CESM2(WACCM6ma) rather than January.

Overall, although there are differences in the peak cooling and longevity of the volcanic cooling among all models, (which are further dependent on the anomaly reference period), there do not appear to be consistent differences between the models that prescribed the optical properties vs. those that used $SO_2$ emissions. For example, the comparison between the model-simulated cooling and the tree-ring records shows a closer match for UKESM1 for 1257 Mt. Samalas, but this is not the case for the two other interactive models (MRI-ESM2 and CESM2(WACCM6ma)), and a smaller cooling is found in IPSL-CM6A-LR which prescribed the optical properties. This reduced cooling does demonstrate however, that there is reconciliation between tree-ring records and model simulated cooling in this latest generation of climate models regardless of how the volcanic forcing is implemented, but that this is also dependent on internal variability. For example, the spread among UKESM1 and MPI-ESM1-2-LR ensemble members is comparable to differences between the models. For 1815 Mt. Tambora, UKESM1 and MRI-ESM2 (both using $SO_2$ emissions) have the smallest simulated cooling but all models except the second MPI-ESM1-2-LR ensemble member and CESM(WACCM6ma) show a cooling that is comparable to the range of tree-ring reconstructions. For the 1458 eruption, the results are also complicated by the different season for this eruption in the two volcanic forcing implementations.

Differences are therefore due not only to the way that volcanic eruptions are implemented in the models ($SO_2$ emissions vs. prescribed) but to other model specifics, for example, in how the optical properties are initially applied in each model for those that prescribe them, how the SAOD is translated into radiative forcing by the model, dependent on the model's radiation scheme as well as other factors such as cloud coverage and insolation, how the forcing translates into the temperature response, dependent on the model's climate sensitivity, as well as on internal variability. Consequently, to explore the role of the two different ways of simulating eruptions independently from differences due to different models, we have run additional simulations of the three largest eruptions in UKESM1 using both approaches. These results are presented in the following section.

### 3.3 1257 Mt. Samalas, 1458 unidentified, and 1815 Mt. Tambora UKESM1 case studies

Using UKESM1 in its preindustrial configuration, we ran two sets of ensemble simulations for each eruption (Sect. 2.1.1), one using $SO_2$ emissions as in our transient simulations, and the other using the aerosol optical properties derived from EVA(2k) (Table 2). For 1458, we conducted eruption scenarios for both January (to match the season used in EVA(2k)) and July (matching how the eruption is simulated in the UKESM1 transient runs). Because the eruptions are simulated in isolation and are therefore not impacted by previous events and because it is also cleaner to derive the volcanic anomaly





(compared with proxy reconstructions) due to having simulations with and without the eruptions, here we do not focus on a
dedicated comparison to the tree-ring records, but rather focus on the differences between the two sets of model ensembles.

**Table 2. UKESM1 case study simulations**

| Eruption | Eruption season | No. of ensemble members – emissions driven | No. of ensemble members – prescribed optical properties |
|---|---|---|---|
| 1257 Mt. Samalas | July | 9 | 9 |
| 1458 Unidentified | Jan/July | 18 (9 Jan, 9 Jul cases) | 9 (Jan only) |
| 1815 Mt. Tambora | April | 9 | 9 |
| Control | No eruption | 9 | 9 |

The global mean, NH mean (20-90°N) and zonal mean SAOD for the three eruptions in the $SO_2$ emissions UKESM1
configuration versus that from EVA(2k) are shown in Fig. 5. Most noticeably, the EVA(2k) dataset prescribes a much more
even spread of aerosol across both hemispheres, leading to similar values between the global and NH averages, whereas
UKESM1 has much stronger SAOD in the southern hemisphere (SH) for Samalas and Tambora (both situated at 8°S) and a
stronger SAOD in the NH for the January 1458 scenario (simulated at 0°N; global and NH averages are similar due to the
concentrated equatorial peak). For the July 1458 scenario (equivalent to how 1458 is simulated in the transient run), the
SAOD in UKESM1 is more evenly spread between the hemispheres, still with a strong tropical peak (not shown, but
equivalent to the transient SAOD in Fig. S7). These features are consistent across all nine ensemble members. As indicated
in the transient runs, the SAOD in the UKESM1 simulations has higher peak values (zonal mean values are greater than 2;
Fig. 5) but decays more quickly than in EVA(2k). Small differences in the actual SAOD seen by UKESM1 compared to the
EVA(2k) values are expected due to the interpolation of the optical properties onto the UKESM1 radiation bands and the
additional very small background stratospheric contribution due to non-volcanic sources present in UKESM1 (Sect. 2.1.1).
These values reflect the magnitude and distributions presented in Fig. 5 (not shown).



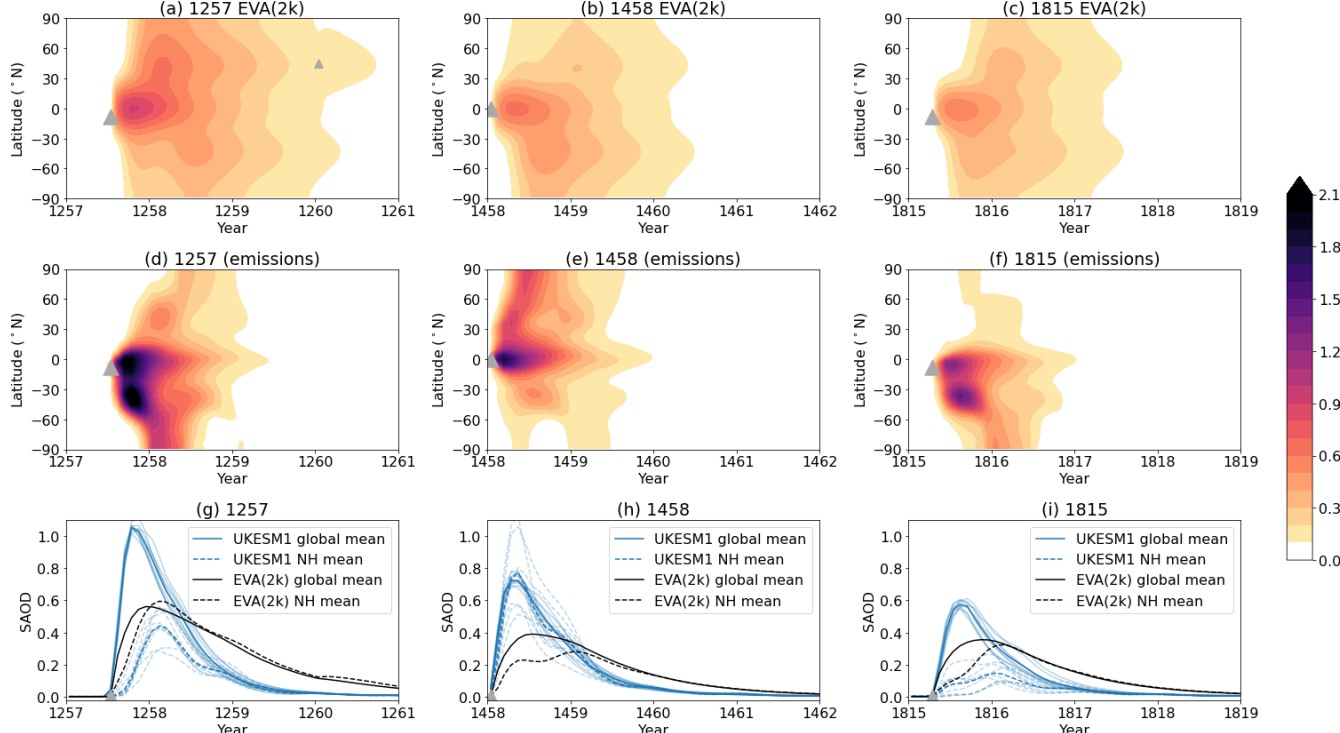

**Figure 5: Zonal mean SAOD in EVA(2k) for the three largest eruptions (a-c) and in the UKESM1 SO₂ emissions-driven**
**simulations (d-f) (ensemble mean), and global mean (solid lines) and NH (20-90°N; dashed lines) SAOD (g-i). The bolder lines**
**mark the UKESM1 ensemble mean and the lighter lines the nine ensemble members. In (e), the ensemble mean is shown for the**
**January simulations for direct comparison with EVA(2K). Grey triangles mark the eruption date and latitude.**





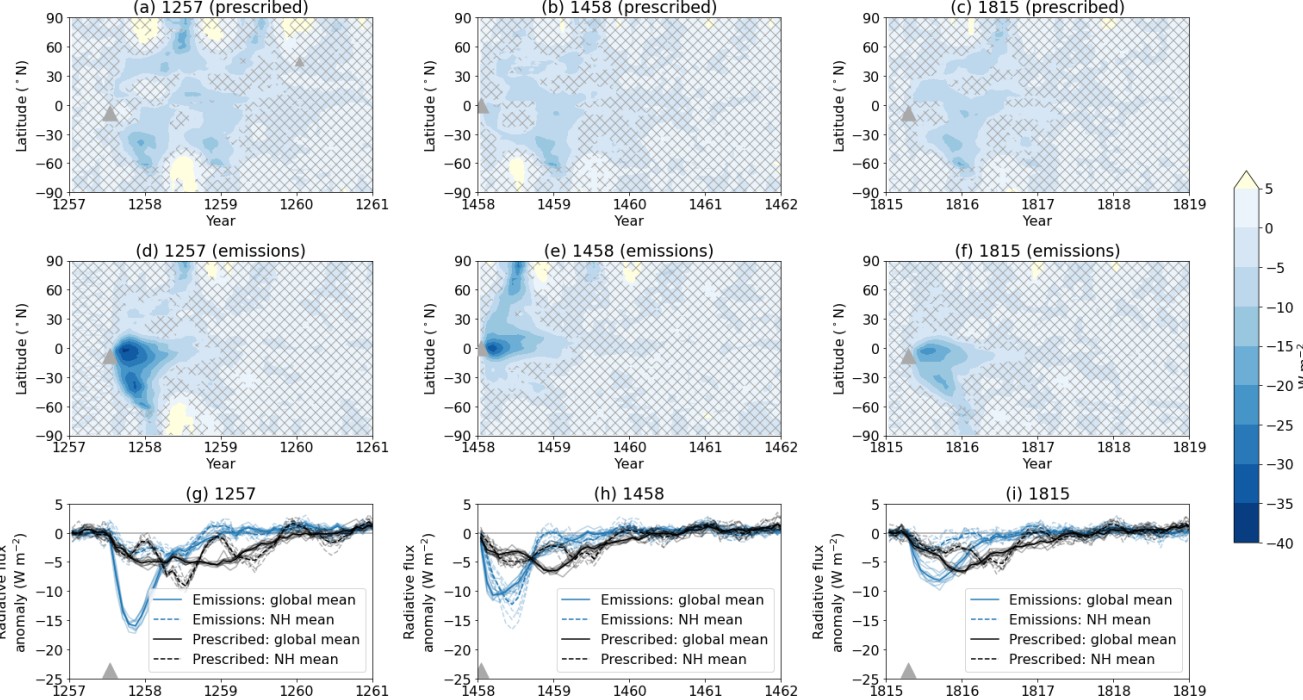

**Figure 6: Zonal mean all-sky top-of-the-atmosphere total radiative flux anomalies from the prescribed runs (a-c) and from the SO₂ emissions-driven runs (d-f) (ensemble mean) and global mean (solid lines) and NH (20-90°N; dashed lines) anomalies (g-i). The bolder lines mark the UKESM1 ensemble mean and the lighter lines the nine ensemble members. The total flux is calculated from the anomaly in longwave + shortwave outgoing radiation multiplied by -1 to show the downward change (incoming solar radiation cancels between the perturbed and control runs). Zonal mean anomalies that are greater than two standard deviations from the nine controls are unhatched. Grey triangles mark the eruption date and latitude.**

Top-of-the-atmosphere total radiative flux anomalies (Fig. 6), mirror the different SAOD distributions in the two volcanic implementation approaches, with stronger, more localised radiative forcing in the SO₂ emissions simulations. The decomposition of the forcing into its longwave and shortwave components (Figures S12-S14) also reveals further differences between the two approaches, including differences in the balance between longwave and shortwave forcing. For example, in the emission-driven simulations the negative shortwave forcing is initially stronger than the longwave forcing but after about a year after an eruption the two forcings become almost equal and quickly tail off. In contrast, the prescribed simulations have weaker shortwave forcing but slightly higher peak values of positive longwave forcing in the first few months after an eruption, leading to a weaker net forcing (smaller negative values). Several factors may influence the balance between shortwave and longwave forcing, including the size distribution of particles and the altitude and latitude of the aerosol through the seasons. Particle sizes can evolve due to microphysical and chemical processes in the emission-driven simulations but in the prescribed simulation the optical properties were derived from a particle size distribution in which the effective radius was linked to the mass of sulfate via an empirically-based relationship based on 1991 Pinatubo (Toohey et al., 2016a). For Samalas, both sets of simulations show net positive forcing in the SH polar regions that is outside of the





control variability, and for the prescribed simulations, additionally in the NH. The prescribed runs have longer-lived net negative forcing in the NH compared to the SO$_2$ emissions-driven simulations. Ensemble spread in the SO$_2$ emissions-driven

runs is larger than in the prescribed runs because of the interactive aerosol implementation, which allows the aerosol to evolve differently each time depending on the initial conditions.

Zonal mean surface air temperature anomalies are shown in Fig. 7 and show distinctive differences between the two volcanic forcing implementations. The prescribed runs show two bands of cooling either side of the equator with ensemble mean

anomalies of around -0.5 to –1 K, but stronger peak anomalies in the NH (> -2 K). The emissions-driven runs on the other hand, show in general stronger localized cooling, especially for 1257 Samalas and 1458. For both 1257 and 1815, the emissions-driven simulations have smaller NH cooling (both in peak and duration). Consequently, the NH summer land cooling (the tree-ring target) differs between the two ensembles (Fig. 8), with stronger peak cooling in the prescribed simulations for 1257 (which also occurs in the second rather than the first year following the eruption) than the emissions-

driven simulation (-1.7 K vs. -1.1 K; ensemble mean) and for 1815 (-1.2 K vs. -0.6 K; ensemble mean). For 1458, the prescribed and July emissions-driven simulations, in which the eruption occurs 6 months later than in EVA(2k), have similar peak cooling (-1.2 K and –1.3 K). However, in the January emissions-driven simulations, with the same eruption date as EVA(2k), peak cooling is much stronger, with an ensemble mean peak of –2.5 K in the year of the eruption. These differences are statistically significant (p<0.05; see purple squares on Figure 8).




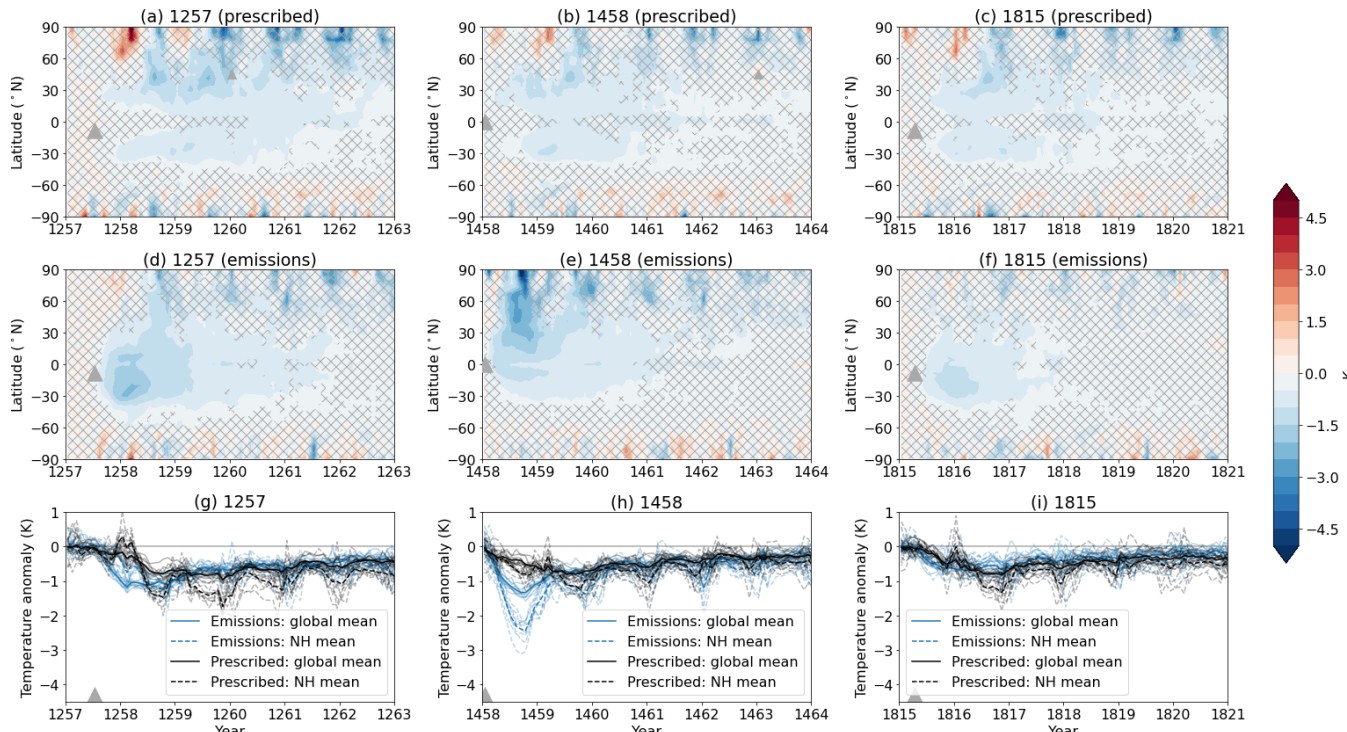

**Figure 7: Zonal mean surface air temperature anomalies in the prescribed simulations (a-c) and the SO₂ emissions-driven simulations (d-e) and global mean (solid lines) and NH (20-90°N; dashed lines) anomalies (g-i). The bolder lines mark the UKESM1 ensemble mean and the lighter lines the nine ensemble members. Anomalies that are greater than two standard deviations from the nine controls are unhatched. Grey triangles mark the eruption date and latitude.**

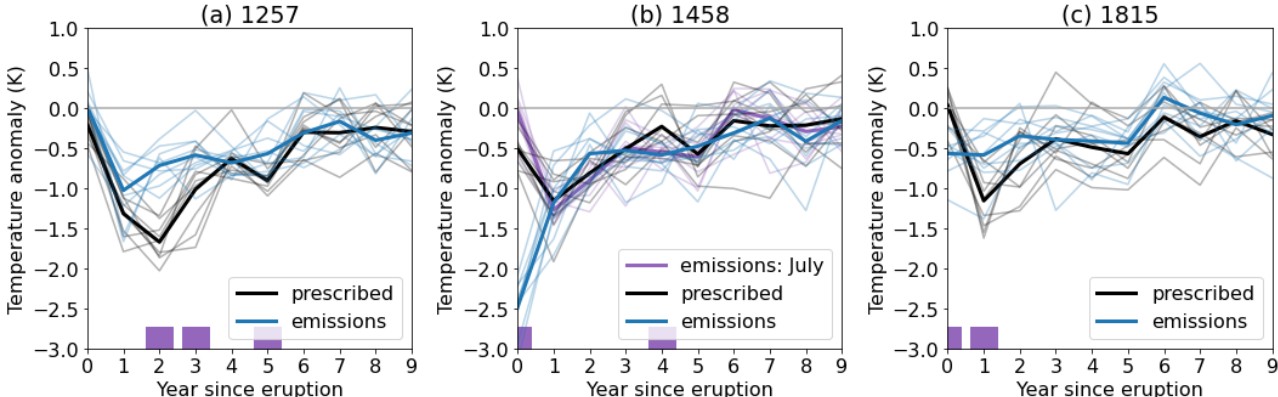

**Figure 8: NH summer land surface air temperature anomalies in the prescribed (black) and SO₂ emissions-driven (blue) simulations. In (b), the additional July emissions-driven case studies are shown in purple. The ensemble mean values are shown in the thicker line. Purple squares indicate where the prescribed and emissions-driven ensembles (nine members) are statistically significantly different (p<0.05) using the two-sided Wilcoxon test (e.g., Zanchettin et al., 2022).**



## 4. Discussion

### 4.1 Last millennium transient simulations

In this study we explore the impact of simulating eruptions during the last millennium using $SO_2$ emissions, instead of
directly prescribing the optical properties using the latest dataset, EVA(2k). We examined the NH summer surface cooling
following eruptions in six climate models that have run the PMIP4 last millennium experiment, which includes a mix of the
two volcanic forcing implementations (Table 1). Superposed epoch analysis (SEA) of the 17 large-magnitude eruptions
simulated between 1250 and 1850, revealed similar average peak cooling across all models (except for
CESM2(WACCM6ma)), despite different forcing implementations, and a stronger cooling compared to four tree-ring
records. This suggests that the estimated $SO_2$ emissions of these eruptions and volcanic forcing, and/or the climate sensitivity
to the volcanic forcing in the models remains too high, or that the proxy records underestimate the large-scale cooling. Lücke
et al. (2023) recently showed that volcanic forcing uncertainty, based on new ensembles of the EVA(2k) reconstruction that
account for emission uncertainty and dating uncertainty, can account for some of the discrepancy between model-simulated
and tree-ring derived cooling for several eruptions such as 1640, 1695, 1783 Laki and 1815 Mt. Tambora. Results of the
SEA from reconstructions will be sensitive to errors of estimated eruption dates, while the simulated eruptions dates are
known and removing just one or two events can completely change the comparison. SEA for only large-magnitude tropical
eruptions simulated in the same seasons in the models, resulted in a closer match between models and tree-rings, especially
for IPSL-CM6A-LR and the stronger Guillet2017 reconstruction (Fig. 3c).

For several individual eruptions, there were clear differences between the models that are not associated with the way the
eruptions were simulated. For example, both UKESM1 ($SO_2$ emissions) and IPSL-CM6A-LR (prescribed optical properties)
simulate a smaller cooling following 1257 Mt. Samalas that compares well to tree-rings. This highlights that further model
differences are at play, such as how the optical properties are translated into radiative forcing including how the prescribed
extinction is translated onto each model's radiation bands and how the tropopause was accounted for in the models that
prescribe the optical properties (see Zanchettin et al., 2022 for further discussion). This was recently demonstrated by
Villamayor et al. (2023) who showed that the shortwave radiative anomalies in IPSL-CM6A-LR, MPI-ESM1-2-LR and
MIROC-ES2L following the 1458 eruption varied considerably, despite all using the EVA(2k) optical properties, with IPSL-
CM6A-LR showing the smallest NH anomalies and MIROC-ES2L showing a different spatial pattern compared with IPSL-
CM6A-LR and MPI-ESM1-2-LR (see their Figure S2). These results are also in line with results from the Volcanic Forcings
Model Intercomparison Project (VolMIP) where for the 1991 eruption of Mt. Pinatubo, in which the same optical properties
were prescribed among six models, regional temperature responses still differed (Zanchettin et al., 2022). Overall, there are
clear differences in the simulated SAOD across the three models that used interactive aerosol (UKESM1,
CESM2(WACCM6ma) and MRI-ESM2), despite the same $SO_2$ emissions, with differences in peak magnitude, hemispheric
dispersion, and decay timescales (Figures 1, S5-8). Differences are due to the different model aerosol schemes as well as





differences in the injection altitude of the emissions (Sect. 2.1). CESM2(WACCM6ma) consistently shows an extremely strong forcing and corresponding cooling for the largest eruptions, likely related to the widths of the aerosol modes in this model's aerosol scheme and a higher climate sensitivity (e.g., Chylek et al., 2020). MRI-ESM2 simulates a much stronger cooling following 1257 Mt. Samalas in comparison to MPI-ESM1-2-LR and MIROC-ES2L, but a smaller response for 1815 Mt. Tambora due to weaker aerosol forcing in the NH (Fig. S8). MRI-ESM2 also shows high SAOD values in the SH polar

vortices, suggesting a strong confinement of aerosol in this region. For Tambora, all models except CESM(WACCM6ma) simulate cooling that matches one of the tree-ring records although the spread between ensemble members from MPI-ESM1-2-LR (Fig. 4f), further demonstrates the importance of meteorological variability and large-scale variability such as ENSO in reconciling models and proxy or instrumental records (e.g., Lehner et al., 2016; Duan et al., 2019; Timmreck et al., 2021). Other model differences likely contributing to the range of temperature responses to these last millennium eruptions include

model resolution and other processes/features that might affect the eventual climate impact (e.g., cloud distribution, overall model climate sensitivity).

## 4.2 UKESM1 case study simulations

Due to the range in model responses for the large-magnitude eruptions, our attention focused on using UKESM1, allowing a direct comparison between the two volcanic forcing implementations in the same model. Our results show that the spatial

pattern of volcanic forcing is very different in UKESM1 between the prescribed set-up and that in which $SO_2$ emissions are used (Figure 6). In UKESM1, we found that the different volcanic forcing implementation strongly affects the resulting NH summer anomalies, with reduced cooling in the emissions-driven scenarios for 1257 Samalas and 1815 Tambora, and consequently a better comparison to tree-rings because of a reduction in the NH radiative forcing.

For 1257 Mt. Samalas and 1815 Mt. Tambora, UKESM1 simulates a stronger dispersal of aerosol to the SH, in contrast to the more evenly distributed hemispheric forcing in the EVA(2k) dataset. This suggests that a different spatial dispersion of the aerosol, and a more asymmetric forcing across the hemispheres from tropical eruptions could reconcile discrepancies between model-simulated and tree-ring derived cooling, when compared to the more global distributions as previously suggested (Toohey and Sigl, 2017; Gao et al., 2008; Crowley and Unterman, 2013). These findings are in line with

Timmreck et al. (2021) who suggested that removing the aerosol forcing from the NH extratropics could result in a better match to NH tree-ring records following the 1809 eruption. However, this is not the case for the unidentified eruption in 1458 where our emission-driven simulations led to a stronger cooling than in the prescribed simulations. This is because of a much stronger NH forcing due to the stronger NH branch of the Brewer Dobson Circulation in January, resulting in more aerosol in the NH. In the simulations of this eruption in July (purple line on Figure 8b), the peak NH cooling is comparable

to the prescribed simulations, demonstrating the importance of eruption season, which will also play a role in reconciling discrepancies between models and proxy reconstructions (e.g., Wainman et al., 2023; Stevenson et al., 2017; Stoffel et al., 2015). The first year of summer cooling will be stronger for eruptions that occur before the first growing season in the



eruption year because of the time it takes for the peak aerosol burden to occur, which may then coincide with peak summer
insolation. Superposed epoch analyses which average multiple eruptions and consider only yearly values (either an annual
mean or the summer average) do not account for the effect of eruption season and can therefore mask differences in the
volcanic response.

In UKESM1, the strong hemispheric asymmetry in forcing is strongly affected by the exact eruption latitude within the
tropics, with both the 1257 Samalas and 1815 Tambora eruptions simulated at 8°S leading to more aerosol in the SH (the
1458 eruption was simulated at 0°N). In a sensitivity test for Samalas (not shown), simulating the eruption at 0°N rather than
at 8°S results in a more symmetric distribution of the SAOD across the hemispheres. UKESM1 also has a much stronger
confinement of aerosol to the tropical region compared with the EVA(2k) dataset in which the SAOD is more quickly spread
across the globe, likely a result of an isolated tropical pipe in this model; Bednarz et al. (2023) found that UKESM1 had the
strongest tropical confinement of sulfate aerosol compared to two other models (CESM2(WACCM6) and GISS-E2.1-G) in
sulfur geoengineering experiments. In favour of the much stronger SH forcing for Samalas and Tambora compared to the
EVA(2k) dataset, our results are comparable with simulated AOD presented in Stoffel et al. (2015) (see their supplementary
figures S7-S9) and from MRI-ESM2 (Figure S9). We also found that the 1286 eruption led to almost double the NH summer
cooling in our UKESM1 simulations than 1257 Mt. Samalas (Figure 2b) despite a much smaller $SO_2$ emission (30 Tg vs. 119
Tg), likely a result of it being simulated at 0°N and in January, leading to a much larger NH aerosol burden, and occurring
before the first growing season (giving time for the peak aerosol to form by summer). For tropical eruptions in the satellite
era (post 1979), asymmetric transport of aerosol has been observed following the 1982 eruption of El Chichón (stronger NH
transport, volcano at 17°N) and 1963 eruption of Agung (stronger SH transport, volcano at 8°S) in contrast to the 1991 Mt.
Pinatubo eruption (15°N), which had a more even distribution between the hemispheres (Timmreck et al., 2021; Thomason
et al., 2018). This further highlights the importance of eruption latitude, season and meteorological conditions for the
evolution of the aerosol cloud. However, it should also be noted that to capture the more even distribution of aerosol to both
hemispheres following 1991 Pinatubo, previous model simulations including the UK climate model have had to spread the
emissions between 15°N and the equator (Dhomse et al., 2014; Mills et al., 2017; Mills et al., 2016; Sheng et al., 2015). This
is to account for the initial southward movement of the aerosol that the models do not otherwise capture, suggested to be due
to missing factors such as the meteorological conditions at the time and processes related to the volcanic plume, or because
the initial tropical confinement and then transport in the hemisphere of the eruption is too strong. It is therefore possible that
the more restricted SAOD in UKESM1 for the large eruptions presented here is affected by these factors.  The models also
do not consider the co-emission of ash, volcanic halogens or water vapour, all of which will likely play a role in the
evolution of the sulfate aerosol and climate response (e.g., Wells et al., 2023; Staunton-Sykes et al., 2021; Abdelkader et al.,
2023; Stenchikov et al., 2021).




The 1458 case is further complicated by the relatively minor cooling in the tree-ring records following this eruption (see also e.g., Schneider et al., 2017; Esper et al., 2017). Our emissions-driven case study simulations in which the 1458 eruption is simulated in isolation, suggest a July date is more likely considering the smaller response, which could perhaps be missed by tree-ring records, although no individual ensemble member lacks the cooling signal. On the other hand, the large cooling

simulated following the January simulations seems more unlikely to not have been captured in tree-rings. The role of the earlier eruption, also in the case for the 1809 and 1815 double event, is difficult to disentangle. The tree-ring records are not without their own uncertainty, and different reconstructions show different responses, but given uncertainties in the volcanic forcing (i.e., season and emission magnitude), it may be that the forcing implemented in the models is not what happened in the real world, further complicating model-proxy comparisons.


In addition to uncertainties in the emission magnitude, uncertainty in the volcanic forcing also arises from the unknown height at which the emissions were released. In our emissions-driven simulations all injections occurred between 18 and 20 km, in line with assumptions made in EVA(2k) to be consistent with the emissions height following the 1991 eruption of Mt. Pinatubo. However, the height is also an important driver of the aerosol evolution in determining the initial spread, tropical

confinement, and aerosol size, leading to differences in the forcing (Stoffel et al., 2015; Marshall et al., 2019; Toohey et al., 2019). The QBO phase at the time of eruption will also impact the initial spread and tropical confinement of the aerosol, and the aerosol itself will impact the QBO (Brown et al., 2023), all of which may play a role in the further reconciliation of simulated climate impacts and those from proxy reconstructions.

The way the eruptions are simulated will also affect other climate variables with larger differences for regional changes given differences in the spatial forcing (e.g., Yang et al., 2019). Different distributions of the forcing and in particular more asymmetric forcing between the hemispheres will have implications on precipitation and modes of variability such as the position of the Inter-Tropical Convergence Zone (e.g., Haywood et al., 2013; Colose et al., 2016), the response of ENSO, the North Atlantic Oscillation, the Southern Oscillation and Atlantic Multidecadal variability (e.g., Timmreck et al., 2021; Fang

et al., 2021) the Atlantic Meridional Overturning Circulation (e.g., Pausata et al., 2015) and tropical cyclone activity (Jones et al., 2017; Yang et al., 2019). Yang et al. (2019) also demonstrated that the transient climate sensitivity was different depending on the spatial distribution of the forcing when comparing the responses of 1902 Santa Maria, 1963 Agung and 1991 Mt. Pinatubo, likely a result of different forcing over land and sea and differences in ocean heat uptake. Villamayor et al. (2023), using four of the models here, also found distinct differences in the response of Sahel rainfall depending on the

symmetry of the forcing in the context of tropical versus extratropical eruptions. Our more asymmetric forcing following the tropical eruptions in UKESM1 would likely lead to further differences in the response of Sahel precipitation to tropical eruptions.





### 4.3 Reconciling model-simulated and tree-ring reconstructed NH summer surface cooling

Overall, our results demonstrate that some of the latest generation of climate models do compare well with tree-ring
reconstructions for the largest eruptions, regardless of how the volcanic forcing is implemented. The latest aerosol forcing dataset (Toohey and Sigl, 2017) does account for self-limiting aerosol microphysical effects that reduce the forcing, although not in as much detail or realism as interactive aerosol models, and chemical and dynamical interactions which can affect the aerosol distribution are also not included. Our case study simulations instead show the importance of spatial pattern of the aerosol for the NH forcing. Stoffel et al. (2015) suggested that nonlinear aerosol microphysical effects and eruption season
could largely reconcile discrepancies, and the AOD in their study is also much stronger in the SH. In UKESM1 the spatial pattern of aerosol forcing is very different when prescribing optical properties vs. using $SO_2$ emissions and this leads to a better agreement with proxy reconstructions for Samalas and Tambora when using emissions. However, the story is not so simple when comparing with other models that have run last millennium simulations. For example, IPSL-CM6A-LR has very similar cooling and prescribed optical properties using the more symmetrical EVA(2k) dataset, demonstrating that the
resulting forcing is highly model dependent. For the comparison for Samalas, it should also be noted that for this period, the amount of MXD tree-ring data that is available is substantially less than for the 1450s and early 1800s (Esper et al., 2018), which will affect the fidelity of the reconstructed values in these records.

### 5 Conclusions

We have investigated the impact of volcanic eruptions on the climate of the past millennium, focussing on the period 1250-
1850 and the difference between representing eruptions using $SO_2$ emissions versus prescribing the optical properties. We find that the stratospheric aerosol optical depth (SAOD) from models that use $SO_2$ emissions and simulate aerosol interactively is different from the latest dataset, EVA(eVolv2k) (Toohey and Sigl, 2017) (Figs. 1, 5, S5), which is the recommended forcing for models that do not have or do not use interactive stratospheric aerosol schemes. The interactive aerosol models tend to simulate more asymmetric aerosol forcing following several large tropical eruptions, compared with
EVA(eVolv2k), which has a more even hemispheric and simplistic distribution, with differences also in the peak SAOD and its longevity (Figs. 1, S6-9)

Across six climate models, which used a mix of the two methods of volcanic forcing implementation, the average NH summer surface cooling response to 17 large-magnitude eruptions is, however, similar and remains larger than the tree-ring
reconstructed cooling (Fig. 3). However, for several individual eruptions, the models do show different responses and a good comparison with tree-ring records is found for 1257 Mt. Samalas for UKESM1 and IPSL-CM6A-LR, and for 1815 Mt. Tambora for all models except CESM2(WACCM6ma) (Fig. 4). Consequently, no clear dependency of the magnitude of cooling on the way volcanic eruptions were included was identified, although other model differences may be at play.

However, using the same model (UKESM1), we find that using $SO_2$ emissions versus prescribing the optical properties leads to a smaller NH summer cooling that aligns better with tree-ring records for 1257 Mt. Samalas and 1815 Mt. Tambora (Fig. 8). Overall, our results suggest that there can be some reconciliation between model-simulated and tree-ring derived cooling for the largest tropical eruptions, which in UKESM1 results from the more asymmetric aerosol forcing following the eruptions compared with the EVA(eVolv2k) dataset. For eruptions in the historical period (post 1850), the EVA model

continues to produce more symmetric aerosol forcing across the hemispheres compared with the CMIP6 aerosol forcing dataset (Thomason et al., 2018). It is therefore important to further consider what is the most realistic spatial distribution of forcing following tropical eruptions in future simulations of last millennium eruptions.

Future work should focus on further understanding some of the model differences presented here that are not necessarily a

function of how the eruptions are simulated. We also do not explore the role of emissions and dating uncertainty of eruptions, which in addition to uncertainties in the spatial patterns of the forcing and the climate response, will also likely aid in the further reconciliation of model simulated climate variables and those derived from proxy reconstructions (e.g., Lücke et al., 2023; Timmreck et al., 2021). The eruption season is also shown here to play a key role in the NH summer temperature anomalies yet for most eruptions in the last millennium remains unknown. Resolving the season will be crucial

in understanding many climate responses to these past eruptions. Furthermore, this work also highlights the need for developing SH tree-ring reconstructions that may be used to better constrain the magnitude and timing of cooling in the SH and consequently spatial forcing for these eruptions. Differences found in surface temperature response between our $SO_2$ emissions-driven and prescribed simulations illustrate the importance of the exact magnitude and spatial structure of the volcanic forcing and we encourage further studies that explore other climate impacts that may depend more strongly on the

spatial evolution of the aerosol.

**Data availability**

Summary model data for the transient runs can be found in the supplementary excel spreadsheet. UK-ESM1 data will be uploaded to the CEDA archive, available at https://archive.ceda.ac.uk/.

**Author Contribution**

Conceptualization: LRM, AS
Data curation: LRM
Formal Analysis: LRM, APS, LJL
Funding acquisition: AS, NLA, RW, KJA, GCH
Investigation: LRM, AS, APS, NLA, LJL



Methodology: LRM, AS, APS, NLA, RW, KJA, GCH, BJ, BOB, ECB, MK, KY

Supervision: AS

Visualization: LRM

Writing – original draft preparation: LRM

Writing – review & editing: All

**Competing interests**

The authors declare that they have no conflict of interest.

**Acknowledgements**

With thanks to all those involved in the generation of PMIP4 data used in this study, to Johann Jungclaus and Jiang Zhu for the provision of model data, to Ken Carslaw and Piers Forster for involvement in the initial proposal for this project and for
useful comments on this manuscript, and to Till Kuhlbrodt, Alistair Seller, Eddy Robertson, and Doug Kelley for UKESM1 modelling support. LRM, AS, NLA, APS, LJL, GCH, and RB were funded by the Natural Environment Research Council project Vol-Clim (NE/S000887/1). KJA was funded by grants from the US National Oceanographic and Atmospheric Administration (NOAA; NA18OAR4310420) and the US National Science Foundation (NSF; AGS-1803946 and AGS-2102993). This work used the ARCHER UK National Supercomputing Service (2013 to 2021), the NEXCS High-
Performance Computing facility funded by the Natural Environment Research Council and delivered by the Met Office between 2017 and 2021, Monsoon2, a collaborative High-Performance Computing facility funded by the Met Office and the Natural Environment Research Council, and JASMIN, the UK collaborative data analysis facility.

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
