# Peer review of "Last Millennium Volcanic Forcing and Climate Response using SO2 Emissions"

_EGUsphere, 2024_

## Community Comment (CC1)

**Comment:**
Marshall et al., Last Millennium Volcanic Forcing and Climate Response using $SO_2$ Emissions

Dear Lauren and others: This paper is a long awaited and important contribution. It is clearly structured and well written, but I do have few comments on aspects related to the volcanic forcing, the analysis of the climate response, and on the 1458 eruption widely assigned to Kuwae.

I would like to start with a more general comment in the beginning: I always thought that the idea of all MIPs (including PMIP, CMIP) is to explore the range different models respond to the same given forcing. For PMIP3, this was already hampered by having two different volcanic forcing datasets at the same time (Gao et al., 2008, Crowely & Unterman 2012). In this paper you randomized the eruption seasons for unidentified eruptions which are now different to those suggested for past1000 simulations in PMIP4 (Toohey & Sigl 2017). There may be good reasons as you outline in your paper to avoid bias or to explore the role of the season on the climate response specifically, but it makes the comparison across the model simulations more difficult.

Before **discussing the effects of the randomization of the dates** it is important to remind ourselves how these dates have been originally derived. All unidentified eruptions (UE) are based on ice-core records of sulfur or sulfate which are available from both Greenland and Antarctica. State-of-the art records are based on annual-layer counted chronologies, constrained to various extent with historical eruption dates (e.g. in 1362, 1477, 1600) and typically have sub-annual (i.e. nominal monthly to seasonal age resolution, so >4 to 12 samples per year). Toohey & Sigl (2017) made a conservative statement of dating uncertainty in the ice cores of better than ±2 years going back to 500 CE. Work published since has widely confirmed the accuracy of the ice core record for the past millennium using records of lunar eclipses (1100-1300 CE) or links to historic eruptions in Iceland (1300-1500 CE) next to new ice cores analyzed and dated since (Guillett et al., 2023; Plunkett et al., 2023; Stoffel et al., 2022; Abbott et al., 2021; Nardin et al., 2021; Sinnl et al. 2022; Burke et al., 2023). A more realistic estimate of the annual layer ice-core dating since 1200 CE would be +/-1 year at most, with the largest eruption signals in the 13th century and 15th century most likely being dated to the correct year.

In reconstructing volcanic sulfate, we can therefore make time estimates of when the volcanic sulfate deposition at the ice core site exceeded the sulfate deposition from other natural sources. For example, a large volcanic sulfate signal is recorded in the 1450s (let's call it Kuwae, more later) in the annual-layer counted chronologies from NGRIP in 1459.2 (Plummer et al., 2012), NEEM in 1459.1 (Sigl et al., 2013) both from Greenland and in Law Dome in 1458.5 (Plummer et al., 2012) and WDC in 1458.4 (Sigl et al., 2013) both from Antarctica. The seasonal timing information has uncertainties of about 1-3 months based on defining the annual-layer boundaries and the distribution of snowfall throughout the year. Next to the uncertainty in the ice-core layer dating and the seasonal age estimate there is also the uncertainty related to the unknown time lag between a large eruption somewhere in the tropics and the subsequent deposition of aerosols on the polar ice sheets. This time lag is difficult to assess empirically because well-dated large magnitude eruptions are scarce. For Tambora which erupted in April 1815 this time lag was about 6 months (Marshall et al., 2018), and the mean time lag for a number of other large eruptions is in the order of 6 +/-3 months (e.g. Wainman et al., 2024), with both studies using high snow accumulation ice-core sites.

Toohey & Sigl (2017) defined the UE eruption year in the PMIP4 aerosol forcing dataset as the year when the sulfate started to rise, so in the example above the eruption year for Kuwae was set

at 1458. Defining January as the default season was not only for simplicity, but it also made sure that the default eruption date always preceded the start of the volcanic sulfate deposition. A mean lag time between eruption and deposition of 6 months implies that about 50% of the UEs have occurred in the year before the initial volcanic sulfur rise defining the eruption year in Toohey 2017. This has consequences for randomizing dates from UE in the tropics. For example, an UE (or Kuwae) in July 1458 (as suggested in this study) is difficult (though not impossible given the uncertainties discussed above) to reconcile with the start of deposition observed in Antarctica in early summer 1458. But eruptions in October 1453 and October 1809 cannot be reconciled with the start of sulfate deposition in these years (nor with the JJA response in the proxies in 1453). So, in effect, by randomizing the eruption dates for tropical eruption dates without accounting for a time delay you have artificially introduced a seasonal dating bias (towards too young) which might also be reflected in the comparisons with some of the proxy reconstructions.

With the computationally expensive analyses already done, I don't see how to address this in hindsight, but I think a more critical discussion of what you mean with "dating uncertainty" in the eruption dates is warranted which you frequently refer to in the paper without going into much detail. Esper et al. (2017), for example, suggested that the ice-core chronologies are wrong and that the large sulfate signal in 1458 in Antarctica should be dated to 1453 to match the cooling in the Arctic, a suggestion which was widely rejected (Abbott et al., 2021; Nardin et al. 2021; Burke et al., 2023). Or do you mean the uncertainty of the eruption season? These are different aspects and clarity is needed what you mean exactly using this term.

I have another comment regarding your **superposed epoch analysis** (SEA) in Figure 3. In my view, the SEA should isolate the mean idealized climate response to a single large volcanic eruption. It is thus important to provide a representative background period and remove from the individual time segments potential effects from additional large subsequent eruptions, before compositing (i.e. remove all data after 1815 when analyzing 1809). If not, the SEA will underestimate the cooling magnitude and overestimate the persistency in the model and proxy responses (see e.g. Büntgen et al., 2020, their Figure 6 and Supplementary Table S3). You have adopted a volcano-free reference period for the SEA of the three largest eruptions in your Supplementary figure, but not for the other large eruptions included in your Figure 3 which also include some prominent clusters next to 1453/58, 1809/15 e.g. those in 1595/1600 and 1831/1835. The clustering of these likely contributes to the tailing and produces the secondary cooling minima visible in year +6.

Finally, a few words on **Kuwae**: The mid-15$^{th}$ century eruption of Kuwae, Vanuatu has for a long time been linked to the exceptionally large volcanic sulfate signal now dated to 1458 (Gao et al., 2006; and references therein; Newhall et al., 2018). I have no objections to being very conservative and calling the 1450s signals Unidentified 1453 and Unidentified 1458 as was also suggested by Toohey & Sigl (2017). No tephra has been identified in ice cores to geochemically link the signal to the Kuwae eruption. However, recent geochemical work have confirmed the date (as shown above), the correctness of the correlated sulfate signals between Greenland and Antarctica, a purely stratospheric formation of the respective sulfate aerosols (through S-isotope analyses; Burke et al., 2023), and thereby that the sulfur injection estimates are within error comparable to those of other caldera-forming VEI=7 eruptions in 1815 and 1257. The strong asymmetry of sulfate accumulating in the Southern Hemisphere as evidenced by ice cores, together with geochronological, volcanological (volume, caldera-size) and petrologic (sulfur yield) evidence from the source, all suggest that the 1458 ice-core signal remains the strongest

contender for the Kuwae eruption (Burke et al 2023; Ballard 2023, Abbott et al., 2021). Using this combined evidence, Kuwae (Vanuatu) was used for the source and a latitude of 17°S for the latest evolv2k_v4 dataset update submitted to PANGAEA recently (Sigl & Toohey, PANGAEA, in review).

**Specific comments:**

**Figure 2: L. 301-302:** Tree-rings do not reflect eruption years either, but the years of cooling which can be in the year or in the years after an eruption.

**L. 423-24:** I would argue that the tree-ring proxy network is spatially biased towards the Arctic and thus underrepresents the cooling observed in 1458 in large areas of the mid-latitudes (e.g. Central Asia, Europe, N-America). NH summer temperature reductions in 1453 from the maximum latewood density (MXD) record from Esper et al. (2017), in particular, are dominated by tree-ring sites located north of 66°N (50% of all records) in proximity to the Arctic ocean with its seasonal sea-ice cover. Temperature reductions at sites below 66°N in Esper et al., (2017) are much smaller (-1.4 °C rel. to 1961-1990) in 1453 AD and almost as large as in 1458 AD (-1.1 °C rel. to 1961-1990). This is also reflected in proxy compilations and monthly reanalysis which assimilated >170 records in the 1450s including most of the tree-ring records discussed here (Valler et al., 2024) summarized in Figures 1 and 2 (below) showing large scale cooling over Northern Hemisphere land areas in the consecutive summers of 1453/1454 and 1458/1459.

**L. 592-596:** there is, however, no empirical evidence that would suggest that the Samalas 1257 the 1809 or the Tambora 1815 produced strong hemispheric asymmetries of sulfate aerosols. A large network of high-quality ice-core records can be used to benchmark the spatial distribution of sulfate across both hemispheres (Sigl et al., 2014; 2015). For 1458, the spatial spread of sulfate is the opposite to your emission-based simulation, with a much larger sulfate spread over the Southern Hemisphere as could be expected for a large eruption at 17°S such as Kuwae. As it stands now it appears you are giving more credit to an apparently improved match between models and tree rings while largely ignoring existing ice-core observations suggesting otherwise.

**L. 706-707**: The most direct evidence about the spatial distribution of aerosols from past volcanic eruption arguably comes from the ice cores.

[Figure]

[Figure]

**Figure 1:** Composite JJA temperature anomaly for 1458 and 1459 (upper panel) and 1453 and 1454 (lower panel) based on >170 proxy records relative to the time period 1422-1452. (Valler et al., 2024)

[Figure]

**Figure 2:** Proxy records used for the climate field reconstruction in Figure 1 before data assimilation (Valler et al., 2024)

**References:**

Abbott, P. M., Plunkett, G., Corona, C., Chellman, N. J., McConnell, J. R., Pilcher, J. R., Stoffel, M., and Sigl, M.: Cryptotephra from the Icelandic Veiðivötn 1477 CE eruption in a Greenland ice core: confirming the dating of volcanic events in the 1450s CE and assessing the eruption's climatic impact, Clim. Past, 17, 565-585, 2021.

Ballard, C., Bedford, S., Cronin, S. J., and Stern, S.: Evidence at source for the mid-fifteenth century eruption of Kuwae, Vanuatu, Journal of Applied Volcanology, 12, 12, 2023.

Büntgen, U., Arseneault, D., Boucher, É., Churakova, O. V., Gennaretti, F., Crivellaro, A., Hughes, M. K., Kirdyanov, A. V., Klippel, L., Krusic, P. J., Linderholm, H. W., Ljungqvist, F. C., Ludescher, J., McCormick, M., Myglan, V. S., Nicolussi, K., et al.: Prominent role of volcanism in Common Era climate variability and human history, Dendrochronologia, 64, 125757, 2020.

Burke, A., Innes, H. M., Crick, L., Anchukaitis, K., Byrne, M., Hutchison, W., McConnell, J. R., Moore, K., Rae, J., Sigl, M., and Wilson, R.: High sensitivity of summer temperatures to stratospheric sulfur loading from volcanoes in the Northern Hemisphere, P Natl Acad Sci USA, 2023.

Gao, C. C., Robock, A., Self, S., Witter, J. B., Steffenson, J. P., Clausen, H. B., Siggaard-Andersen, M. L., Johnsen, S., Mayewski, P. A., and Ammann, C.: The 1452 or 1453 AD Kuwae eruption signal derived from multiple ice core records: Greatest volcanic sulfate event of the past 700 years, J Geophys Res-Atmos, 111, 2006.

Guillet, S., Corona, C., Oppenheimer, C., Lavigne, F., Khodri, M., Ludlow, F., Sigl, M., Toohey, M., Atkins, P. S., Yang, Z., Muranaka, T., Horikawa, N., and Stoffel, M.: Lunar eclipses illuminate timing and climate impact of medieval volcanism, Nature, 616, 90-95, 2023.

Marshall, L., Schmidt, A., Toohey, M., Carslaw, K. S., Mann, G. W., Sigl, M., Khodri, M., Timmreck, C., Zanchettin, D., Ball, W., Bekki, S., Brooke, J. S. A., Dhomse, S., Johnson, C., Lamarque, J. F., LeGrande, A., Mills, M. J., Niemeier, U., Poulain, V., Robock, A., Rozanov, E., Stenke, A., Sukhodolov, T., Tilmes, S., Tsigaridis, K., and Tummon, F.: Multi-model comparison of the volcanic sulfate deposition from the 1815 eruption of Mt. Tambora, Atmos. Chem. Phys. 2018.

Marshall, L., Schmidt, A., Toohey, M., Carslaw, K. S., Mann, G. W., Sigl, M., Khodri, M., Timmreck, C., Zanchettin, D., Ball, W. T., Bekki, S., Brooke, J. S. A., Dhomse, S., Johnson, C., Lamarque, J. F., LeGrande, A. N., Mills, M. J., Niemeier, U., Pope, J. O., Poulain, V., Robock, A., et al.: Multi-model comparison of the volcanic sulfate deposition from the 1815 eruption of Mt. Tambora, Atmos. Chem. Phys., 18, 2307-2328, 2018.

Nardin, R., Severi, M., Amore, A., Becagli, S., Burgay, F., Caiazzo, L., Ciardini, V., Dreossi, G., Frezzotti, M., Hong, S. B., Khan, I., Narcisi, B. M., Proposito, M., Scarchilli, C., Selmo, E., Spolaor, A., Stenni, B., and Traversi, R.: Dating of the GV7 East Antarctic ice core by high-resolution chemical records and focus on the accumulation rate variability in the last millennium, Clim Past, 17, 2073-2089, 2021.

Newhall, C., Self, S., and Robock, A.: Anticipating future Volcanic Explosivity Index (VEI) 7 eruptions and their chilling impacts, Geosphere, 14, 572-603, 2018.

Plummer, C. T., Curran, M. A. J., van Ommen, T. D., Rasmussen, S. O., Moy, A. D., Vance, T. R., Clausen, H. B., Vinther, B. M., and Mayewski, P. A.: An independently dated 2000-yr volcanic record from Law Dome, East Antarctica, including a new perspective on the dating of the 1450s CE eruption of Kuwae, Vanuatu, Clim Past, 8, 1929-1940, 2012.

Plunkett, G., Sigl, M., McConnell, J. R., Pilcher, J. R., and Chellman, N. J.: The significance of volcanic ash in Greenland ice cores during the Common Era, Quaternary Sci Rev, 301, 107936, 2023.

Sigl, M., McConnell, J. R., Layman, L., Maselli, O., McGwire, K., Pasteris, D., Dahl-Jensen, D., Steffensen, J. P., Vinther, B., Edwards, R., Mulvaney, R., and Kipfstuhl, S.: A new bipolar ice core record of volcanism from WAIS Divide and NEEM and implications for climate forcing of the last 2000 years, J Geophys Res-Atmos, 118, 1151-1169, 2013.

Sigl, M., McConnell, J. R., Toohey, M., Curran, M., Das, S. B., Edwards, R., Isaksson, E., Kawamura, K., Kipfstuhl, S., Kruger, K., Layman, L., Maselli, O. J., Motizuki, Y., Motoyama, H., Pasteris, D. R., and Severi, M.: Insights from Antarctica on volcanic forcing during the Common Era, Nat Clim Change, 4, 693-697, 2014.

Sigl, M., Winstrup, M., McConnell, J. R., Welten, K. C., Plunkett, G., Ludlow, F., Büntgen, U., Caffee, M., Chellman, N., Dahl-Jensen, D., Fischer, H., Kipfstuhl, S., Kostick, C., et al..: Timing and climate forcing of volcanic eruptions for the past 2,500 years, Nature, 523, 543-549, 2015.

Sigl, M. & Toohey M.: Volcanic stratospheric sulfur injections from 500 BCE to 1900 CE, eVolv2k_version4, PANGAEA, submitted.

Stoffel, M., Corona, C., Ludlow, F., Sigl, M., Huhtamaa, H., Garnier, E., Helama, S., Guillet, S., Crampsie, A., Kleemann, K., Camenisch, C., McConnell, J., and Gao, C.: Climatic, weather, and socio-economic conditions corresponding to the mid-17th-century eruption cluster, Clim. Past, 18, 1083-1108, 2022.

Toohey, M. and Sigl, M.: Volcanic stratospheric sulfur injections and aerosol optical depth from 500 BCE to 1900 CE, Earth System Science Data, 9, 809-831, 2017.

Valler, V., Franke, J., Brugnara, Y., Samakinwa, E., Hand, R., Lundstad, E., Burgdorf, A.-M., Lipfert, L., Friedman, A. R., and Brönnimann, S.: ModE-RA: a global monthly paleo-reanalysis of the modern era 1421 to 2008, Sci Data, 11, 36, 2024.

Wainman, L., Marshall, L. R., and Schmidt, A.: Utilising a multi-proxy to model comparison to constrain the season and regionally heterogeneous impacts of the Mt Samalas 1257 eruption, Clim. Past, 20, 951-968, 2024.

---

## Author Comment (AC1)

**Replies to reviewers Marshall et al. 2024**

We are very grateful to Alan Robock, Michael Sigl and an anonymous reviewer for their comments and insights on our paper. Please find in blue our replies below.

**Reviewer 1.**

This paper presents a lot of work and should be published. However, there are so many potential sources of error that it seems that definitive conclusions are hard to make. The paper recommends more study, but it is not clear whether further study will be able to unravel the problems. Some primary ones are chaos in the climate system, impacts of unknown El Niño and La Niña at the time of the eruption, seasonal timing of the growth cycle of the tree rings vs. the volcanic forcing, the latitude of the eruption, and the subsequent latitudinal transport. Some of the figures could be made clearer (see below).

Many thanks for your comments. We agree that a lot of uncertainty still exists in this field but here we focussed on one aspect – the imposition of forcing, which we show has an important effect. There is certainly a lot left to explore in our datasets as suggested by the next comment (e.g. impacts on ENSO, monsoons) but this would be beyond the scope of this initial paper. We have added to the conclusions:

*"This work has highlighted the importance of internal variability which could be addressed by further ensemble member simulations, as well as investigating the role of ENSO at the time of the eruptions. ... we encourage further studies that explore other climate impacts for example on ENSO, summer monsoon precipitation and winter warming, that may depend more strongly on the spatial evolution of the aerosol."*

The analysis focuses on the UKESM model and summer temperature response, presumably because of tree ring proxy data. But why not all the other impacts, including El Niño, summer monsoon precipitation, and winter warming? Still, it would be nice to evaluate the model for these impacts for the modern period, including for surface temperature, so that we can have confidence that the model does a good job in simulating the climate response to volcanic eruptions.

We decided to focus on the summer temperature response as it is one of the main targets when it comes to assessing the climatic effects of volcanic eruptions and although it would be interesting to examine other impacts, this would make the current paper overly long, unfocussed and complicated. We have, however, suggested these further studies in the conclusions (see text above).

The aerosol model has been previously evaluated for modern eruptions (Dhomse et al., 2020) and a comparison to SAOD observations for Pinatubo is shown in Supplementary Figure 1. A full evaluation of the modern timeseries is beyond the scope of this paper and further complicated by the fact that the runs are free-running (i.e. we did not choose the specific ENSO and QBO phases for the eruptions in the modern era).

How do you account for the possible presence of La Niña or El Niño at the same time of the volcanic eruptions you are using to compare to the simulation, both in the real world and the model simulations? For example, we know that the actual 1982 El Chichón response would have been much cooler without the simultaneous El Niño, and that for 1991 Pinatubo it would have been somewhat cooler. (See Soden, Brian J., Richard T. Wetherald, Georgiy L. Stenchikov, and Alan Robock, 2002: Global cooling following the eruption of Mt. Pinatubo: A test of climate feedback by water vapor. Science, 296, 727-730.) El Niño and La Niña are random in the observational record, but I don't think you have data on them at the time of the

volcanic eruptions.  In models, we have found that the probability of an El Niño the next winter is higher with a tropical eruption, but does the model you used respond this way?

As the simulations are free-running in which the meteorological conditions evolve interactively the phases at the time of each eruption are random. A preliminary examination of ENSO (DJF anomaly in year after eruption) in the UKESM simulations suggested a shift to La Nina type conditions following the big three eruptions (see table below), but we suggest a dedicated study be conducted to examine these differences more robustly. The role of ENSO is also hard to evaluate given the different eruption properties (e.g. latitude and $SO_2$ magnitude) and would be better examined following dedicated experiments where only the ENSO phase was changed.

**Table 1:** DJF ENSO in year after eruption in the three UKESM1 ensemble members

| Year | r1 | r2 | r3 |
|---|---|---|---|
| 1257 | La Nina | Neutral | La Nina |
| 1276 | La Nina | Neutral | El Nino |
| 1286 | El Nino | El Nino | La Nina |
| 1345 | Neutral | El Nino | Neutral |
| 1453 | Neutral | El Nino | La Nina |
| 1458 | La Nina | La Nina | La Nina |
| 1477 | El Nino | La Nina | Neutral |
| 1585 | Neutral | Neutral | Neutral |
| 1595 | Neutral | El Nino | La Nina |
| 1600 | El Nino | La Nina | La Nina |
| 1640 | La Nina | Neutral | La Nina |
| 1695 | El Nino | La Nina | El Nino |
| 1783 | El Nino | El Nino | Neutral |
| 1809 | La Nina | La Nina | La Nina |
| 1815 | Neutral | La Nina | La Nina |
| 1831 | El Nino | Neutral | La Nina |
| 1835 | Neutral | Neutral | La Nina |

I find the colors of the triangles in Fig. 1 hard to interpret.  Many look the same and the Southern Hemisphere colors are very similar.  I recommend a much wider range of distinct colors, or actually plotting the latitude on the figure using the scale on the right as the axis. You could then use open circles for the eruptions so as not to hide the other lines.  And please give a scale for the size of circles or triangles.  "Size" is not defined.  Is it width of the symbol, or its area that represents the emissions?  Also the lines for the different data sets are hard to distinguish.  Why are some thinner and some dashed.  Make them thicker and use primary colors – red, green, and blue – to make them distinct.

We have redesigned this figure using a new latitude colour scheme with fewer distinct categories, which hopefully makes this clearer. The size - width and area - is proportional to the emission magnitude and we have added a scale to the legend to clarify this.

The lines are hard to distinguish in places given that for many of the eruptions the SAOD is similar (a result in itself). The EVA(2k) line is dashed to show the comparison with MPI as it

is very similar (except for 1257 and Laki). We have updated the colours and have thickened the lines in all cases.

Why is 1458 Undefined? I thought it was the accepted date for Kuwae. Or are you sure Kuwae was 1452?

This eruption was originally unidentified in the eVolv2k dataset used at the time of these simulations (Toohey and Sigl, 2017) and given sufficient uncertainty we retain the unidentified classification in this work (see also discussion from Michael Sigl below). We have added to the paper:

*"The 1458 eruption has recently been reattributed to Kuwae in the latest eVolv2k dataset (v4) but given sufficient uncertainty we retain the unidentified classification in this work."*

How do you know the latitude of the unidentified eruptions? From the relative depositions in Greenland and Antarctica? How do you account for varying transport strengths both from random climate variation or forced circulation from the eruptions?

The latitudes were determined based on the presence or absence of deposition over each ice sheet as outlined in Toohey and Sigl (2017) – either at the equator or at 45 degrees N/S, rather than assuming a more specific location due to deposition ratio, which would carry further uncertainty given the influence of season and natural variability (as well as other eruption source parameters such as the $SO_2$ emission and injection altitude). Varying transport strength is included due to the free-running nature of our simulations. Its influence can only be determined from multiple ensemble members.

The colors of the lines in Fig. 2 are hard to distinguish. CESM and IPSL are very similar, and the UKESM colors are similar. Make them distinct. The color distinction of the eruption latitude is still a problem, like in Fig. 1. Plot the latitudes as values in the vertical.

Again, in Fig. 9 all the lines have very similar colors. Use bright red, green, and blue to distinguish them.

We have updated all figures that present results from multiple models (figures 1-4) using a different colour scheme and thicker lines. However, due to the number of simulations included and by trying to keep the colours as accessible as possible, it is impossible to have complete separation. However, we believe that all major conclusions presented in the paper can be seen in these figures.

Are you concerned that additional diffuse radiation after volcanic eruptions would enhance tree growth, making what you interpret as pure radiative volcanic signal actually also a signal of diffuse radiation "fertilization?" (See Robock, Alan, 2005: Cooling following large volcanic eruptions corrected for the effect of diffuse radiation on tree rings. Geophys. Res. Lett., 32, L06702, doi:10.1029/ 2004GL022116.) This would lower the tree ring response, making the tree rings an imperfect measure of the temperature.

It is an interesting concept that adds to the uncertainty, however there is currently no real evidence for this effect (mature trees in cold high latitude environments are probably not carbon/assimilation limited) - https://agupubs.onlinelibrary.wiley.com/doi/full/10.1029/2003GB002076, and we have therefore not added anything further to the manuscript.

You mention special treatment of the forcing from the Laki eruption, because we have some observations. But what climate response did you get from it? It is a great illustration, at least on a continental scale, that chaotic weather can cause warming over Europe the

summer of 1783 rather than the expected cooling. How do you address this in your interpretations of the responses to the other eruptions? (See Zambri, Brian, Alan Robock, Michael J. Mills, and Anja Schmidt, 2019b: Modeling the 1783–1784 Laki eruption in Iceland, Part II: Climate impacts. J. Geophys. Res. Atmos., 124, 6770-6790, doi:10.1029/2018JD029554.)

We think that a full evaluation of the response to Laki will distract from the main messages in the paper and will be a focus of a future study. However, since this eruption is highlighted in the main text as showing differences between models, we have added a plot to the Supplementary which shows detailed panels for the 17 large-magnitude eruptions included in the SEA:

[Figure]

Regarding the role of natural variability, we have added the following text about the spread among ensemble members:

*"There is also spread among ensemble members with cooling differing by more than 1 K across the 9 members for all three eruptions and for each method of simulation. This ensemble range is of the same order of magnitude as the differences between the different models (except for CESM(WACCM6ma)) shown in Fig. 4, further demonstrating that model differences may be a result of internal variability."*

**Reviewer 2.**

This paper addresses both model-to-model uncertainty and volcanic forcing uncertainty in the last millennium. The paper is well written, presents interesting results regarding both large model and forcing uncertainty and is a nice contribution to the scientific literature. I suggest it is accepted subject to the suggestions below.

Thank you for your comments.

Major comments:

Figure 1: eruption colorscale is hard to distinguish in the purple colours south of the equator, and are the triangles linearly related to the size of the eruption?

We have updated all figures that compare the results of multiple models with a revised colour scheme and thicker lines. We have also added a legend for the triangle sizes.

As noted by the authors there is spread among different ensemble members of the same model (e.g. line 398-399). Can the 9 idealised UKESM members be used to estimate the internal variability in each model? Then it would be good to put the model differences in context of the internal variability. I.e. if we consider internal variability how different are the models from each other and from the observations?

A further examination of Figure 8 shows that among the 9 ensemble members, the NH summer cooling can differ by more than 1 K in all cases, which is comparable to the model differences, except for CESM(WACCM6a). We have added the following to the paper:

*"There is also spread among ensemble members with cooling differing by more than 1 K across the 9 members for all three eruptions and for each method of simulation. This ensemble range is of the same order of magnitude as the differences between the different models (except for CESM(WACCM6ma)) shown in Fig. 4, further demonstrating that model differences may be a result of internal variability."*

When referring to the differences between a,b, & c in Figure 3 those cited by the authors are really hard to see by eye in the Figure. Are the models really closer together in panel c compared to a? I don't see this by eye.

We have updated this figure with the new colour scheme and have also re-calculated the SEA using 5-year reference periods not affected by another eruption and with the data from double events removed as suggested by Michael Sigl. The anomalies are slightly stronger compared with the original figure and the shape of the tails slightly changed, however the overall comparisons remain the same. It is the IPSL model which is closer to the tree-rings with a greater spread across the models. We have rephrased the text as follows:

*"Panel (c) shows only the eruptions simulated in the same season, reflecting the fairest comparison between the models. For these eruptions (which excludes 1276, 1453, 1458, 1695, 1809) the models are more separated with IPSL-CM6A-LR and UKESM1 r1 and r3 closer to the reconstructions, with IPSL-CM6A-LR comparing very well with the stronger Guillet2017 record."*

It seems CESM is quite different to UKESM . Do you have the data to add CESM to Figure 5 for comparison? I think this would be very useful.

Since this section is focussed on the UKESM1 simulations we think that this comparison will be a further distraction and that the MRI-ESM2 data should also be added. However, the

zonal SAOD for CESM can be seen in Figure S6 and can be directly compared to that of UKESM1 in Figure S7. For interest, please see below for some further direct comparisons:

[Figure]

**Figure 1.** Global and zonal SAOD for UKESM1 and CESM2(WACCM6ma) for 1257 Samalas

[Figure]

**Figure 2.** Global and zonal SAOD for UKESM1 and CESM2(WACCM6ma) for 1458

[Figure]

**Figure 3.** Global and zonal SAOD for UKESM1 and CESM2(WACCM6ma) for 1815 Tambora

Minor comments:

Line 62, 'However' does not make sense in this context

We have removed the 'however'.

Line 110 should near 'now simulate'

Changed.

Line 315 is this drift or something else? Why is CESM warming?

We do not know for sure but possible reasons include natural variability and the model's climate sensitivity which is too great. A single realization is also insufficient to show that it is a robust feature of the model's simulation of the last millennium. It is also possible that depending on the specific aerosol and chemistry schemes in the other models, the CESM response might be more realistic. CESM is also warmer at other points e.g. ~1580.

We have added to the paper: *"and which could be related to internal variability or the model's climate sensitivity."*

Line 333 - if CESM does does something different to the other models can you state what it is doing here?

It is a much stronger cooling response. Possible explanations include the model's climate sensitivity and the specific aerosol and chemistry scheme. CESM has been shown to perform well for Mt. Pinatubo, so it also cannot be discounted that this is still a possible response.

We have added: *"with CESM2(WACCM6ma) displaying a stronger peak cooling >1.5 K"*

Line 610 - can you put the not shown figure in Supp?

We have added this figure to the Supplementary – Figure S16. A re-examination of this analysis also further demonstrated the importance of the initial conditions – two ensemble simulations were run chosen from end-member scenarios for the 1458 case studies resulting

in stronger NH transport in r1, and stronger SH transport in r2 (but weaker than when the eruption is simulated at 8°S).

[Figure]

**Figure S16:** Zonal mean SAOD in simulations of Samalas at 0 degrees.

We have amended the manuscript as follows:

*"In sensitivity tests for Samalas (Figure S16), simulating the eruption at 0°N rather than at 8°S resulted in weaker hemispheric asymmetry and for one realization (out of two), stronger NH SAOD, further demonstrating the role of initial conditions and internal variability."*

Line 658 - add references for the response of ENSO - there are quite a lot of papers that consider this please cite at least one

Papers to consider:

https://www.nature.com/articles/s41586-023-06447-0

https://www.nature.com/articles/s41561-019-0400-0

Thank you. We have added further references.

**Community comment**

Dear Lauren and others: This paper is a long awaited and important contribution. It is clearly structured and well written, but I do have few comments on aspects related to the volcanic forcing, the analysis of the climate response, and on the 1458 eruption widely assigned to Kuwae.

Many thanks for your detailed comments.

I would like to start with a more general comment in the beginning: I always thought that the idea of all MIPs (including PMIP, CMIP) is to explore the range different models respond to the same given forcing. For PMIP3, this was already hampered by having two different volcanic forcing datasets at the same time (Gao et al., 2008, Crowely & Unterman 2012). In this paper you randomized the eruption seasons for unidentified eruptions which are now different to those suggested for past1000 simulations in PMIP4 (Toohey & Sigl 2017). There may be good reasons as you outline in your paper to avoid bias or to explore the role of the season on the climate response specifically, but it makes the comparison across the model simulations more difficult.

In hindsight keeping to the same seasons would have been helpful in this instance, however our UKESM1 simulations were not conducted as part of PMIP4 and therefore we chose to randomise the eruption seasons and focus on the interactive aerosol simulation, which will allow us to address a host of other research questions in the future. We added the SEA for eruptions simulated in the same season because of this uncertainty, with similar results to when all eruptions were included. We have added a section to the discussion further highlighting this issue – see reply below.

Before discussing the effects of the randomization of the dates it is important to remind ourselves how these dates have been originally derived. All unidentified eruptions (UE) are based on ice-core records of sulfur or sulfate which are available from both Greenland and Antarctica. State-of-the art records are based on annual-layer counted chronologies, constrained to various extent with historical eruption dates (e.g. in 1362, 1477, 1600) and typically have sub-annual (i.e. nominal monthly to seasonal age resolution, so >4 to 12 samples per year). Toohey & Sigl (2017) made a conservative statement of dating uncertainty in the ice cores of better than ±2 years going back to 500 CE. Work published since has widely confirmed the accuracy of the ice core record for the past millennium using records of lunar eclipses (1100-1300 CE) or links to historic eruptions in Iceland (1300-1500 CE) next to new ice cores analyzed and dated since (Guillett et al., 2023; Plunkett et al., 2023; Stoffel et al., 2022; Abbott et al., 2021; Nardin et al., 2021; Sinnl et al. 2022; Burke et al., 2023). A more realistic estimate of the annual layer ice-core dating since 1200 CE would be +/-1 year at most, with the largest eruption signals in the 13th century and 15th century most likely being dated to the correct year.

In reconstructing volcanic sulfate, we can therefore make time estimates of when the volcanic sulfate deposition at the ice core site exceeded the sulfate deposition from other natural sources. For example, a large volcanic sulfate signal is recorded in the 1450s (let's call it Kuwae, more later) in the annual-layer counted chronologies from NGRIP in 1459.2 (Plummer et al., 2012), NEEM in 1459.1 (Sigl et al., 2013) both from Greenland and in Law Dome in 1458.5 (Plummer et al., 2012) and WDC in 1458.4 (Sigl et al., 2013) both from Antarctica. The seasonal timing information has uncertainties of about 1-3 months based on defining the annual-layer boundaries and the distribution of snowfall throughout the year. Next to the uncertainty in the ice-core layer dating and the seasonal age estimate there is also the uncertainty related to the unknown time lag between a large eruption somewhere in

the tropics and the subsequent deposition of aerosols on the polar ice sheets. This time lag is difficult to assess empirically because well-dated large magnitude eruptions are scarce. For Tambora which erupted in April 1815 this time lag was about 6 months (Marshall et al., 2018), and the mean time lag for a number of other large eruptions is in the order of 6 +/-3 months (e.g. Wainman et al., 2024), with both studies using high snow accumulation ice-core sites. Toohey & Sigl (2017) defined the UE eruption year in the PMIP4 aerosol forcing dataset as the year when the sulfate started to rise, so in the example above the eruption year for Kuwae was set at 1458. Defining January as the default season was not only for simplicity, but it also made sure that the default eruption date always preceded the start of the volcanic sulfate deposition. A mean lag time between eruption and deposition of 6 months implies that about 50% of the UEs have occurred in the year before the initial volcanic sulfur rise defining the eruption year in Toohey 2017. This has consequences for randomizing dates from UE in the tropics. For example, an UE (or Kuwae) in July 1458 (as suggested in this study) is difficult (though not impossible given the uncertainties discussed above) to reconcile with the start of deposition observed in Antarctica in early summer 1458. But eruptions in October 1453 and October 1809 cannot be reconciled with the start of sulfate deposition in these years (nor with the JJA response in the proxies in 1453). So, in effect, by randomizing the eruption dates for tropical eruption dates without accounting for a time delay you have artificially introduced a seasonal dating bias (towards too young) which might also be reflected in the comparisons with some of the proxy reconstructions.

With the computationally expensive analyses already done, I don't see how to address this in hindsight, but I think a more critical discussion of what you mean with "dating uncertainty" in the eruption dates is warranted which you frequently refer to in the paper without going into much detail. Esper et al. (2017), for example, suggested that the ice-core chronologies are wrong and that the large sulfate signal in 1458 in Antarctica should be dated to 1453 to match the cooling in the Arctic, a suggestion which was widely rejected (Abbott et al., 2021; Nardin et al. 2021; Burke et al., 2023). Or do you mean the uncertainty of the eruption season? These are different aspects and clarity is needed what you mean exactly using this term.

Thank you for this detailed explanation which has given us a lot to think about! This was something we did not consider when randomising the season. We agree that therefore for some of the eruptions this does introduce a further bias. However, these undated eruptions are not a focus of this study and given all of the other uncertainties (especially related to the time between emission and deposition for the large-magnitude unknown eruptions, which is still based on a limited set of modelling studies) and the fact that our simulations are just one possible realization of reality, we think that the comparisons are still very much of value.

With regard to the use of the phrase 'dating uncertainty' we mean in relation to both the year and the eruption season. Although we acknowledge that the years are generally robust, given that we can't rule out a +/- 1 year uncertainty, we think that highlighting this is still important. When first introducing this term in the introduction, we have clarified that we mean both year and season. We have also reduced the use of 'dating' throughout the paper, instead specifying uncertainties in season and year.

We have added the following text to the discussion regarding the choice of season and its implications:

*"Furthermore, both the comparisons between models and to the proxy reconstructions are complicated by the choice of season in our simulations. Although the season was randomised in the UKESM1 and CESM2(WACCM6ma) simulations to avoid potential biases for future analyses, differences in season will contribute to model differences and to the*

*proxy-model comparisons. Some of the randomised seasons (e.g. October for 1809) are also inconsistent with the timescales of measured sulfate deposition to the ice sheets (i.e. that deposition is measured before the eruption date). Although these eruptions are not a focus of this analysis, this should be considered in any future comparisons, as well as uncertainties in the dating from ice cores. Further work investigating the timescales between eruption and deposition, particularly on how this is dependent on the eruption location, season and emission altitude, is warranted (e.g., Wainman et al., 2024)."*

I have another comment regarding your superposed epoch analysis (SEA) in Figure 3. In my view, the SEA should isolate the mean idealized climate response to a single large volcanic eruption. It is thus important to provide a representative background period and remove from the individual time segments potential effects from additional large subsequent eruptions, before compositing (i.e. remove all data after 1815 when analyzing 1809). If not, the SEA will underestimate the cooling magnitude and overestimate the persistency in the model and proxy responses (see e.g. Büntgen et al., 2020, their Figure 6 and Supplementary Table S3). You have adopted a volcano-free reference period for the SEA of the three largest eruptions in your Supplementary figure, but not for the other large eruptions included in your Figure 3 which also include some prominent clusters next to 1453/58, 1809/15 e.g. those in 1595/1600 and 1831/1835. The clustering of these likely contributes to the tailing and produces the secondary cooling minima visible in year +6.

We have re-done the SEA so that all eruptions are now with reference to a 'clean' background and with the anomalies from a double event removed. However, this is still imperfect given that the removal of data could also bias the recovery to lower anomalies, and determining a true background is also difficult especially given other natural variations in temperature. The details of the reference periods are now included in Table S2. We have added the SEA with the 5-years before to the Supplementary and removed the original discussion regarding the anomalies and double events. The plots are not that different – there are slightly stronger anomalies and slightly reduced persistency, with no change to our conclusions.

The manuscript has been revised as follows:

*"To composite the different eruptions, anomalies are calculated with respect to the five years prior to each eruption, unless another eruption has occurred within this period, in which case the closest 5 years prior to this is taken (see Table S2) and we take year 0 as the year of the eruption (other studies have also considered the year of peak aerosol load or year of peak forcing, e.g., Liu et al. 2022). In the case of volcanic double events, in which two eruptions are closely spaced, the data following the second eruption (if it is a NH or tropical eruption) is removed prior to the averaging (Table S2)."*

Finally, a few words on Kuwae: The mid-15th century eruption of Kuwae, Vanuatu has for a long time been linked to the exceptionally large volcanic sulfate signal now dated to 1458 (Gao et al., 2006; and references therein; Newhall et al., 2018). I have no objections to being very conservative and calling the 1450s signals Unidentified 1453 and Unidentified 1458 as was also suggested by Toohey & Sigl (2017). No tephra has been identified in ice cores to geochemically link the signal to the Kuwae eruption. However, recent geochemical work have confirmed the date (as shown above), the correctness of the correlated sulfate signals between Greenland and Antarctica, a purely stratospheric formation of the respective sulfate aerosols (through S-isotope analyses; Burke et al., 2023), and thereby that the sulfur injection estimates are within error comparable to those of other caldera-forming VEI=7 eruptions in 1815 and 1257. The strong asymmetry of sulfate accumulating in the Southern Hemisphere as evidenced by ice cores, together with geochronological, volcanological

(volume, caldera-size) and petrologic (sulfur yield) evidence from the source, all suggest that the 1458 ice-core signal remains the strongest contender for the Kuwae eruption (Burke et al 2023; Ballard 2023, Abbott et al., 2021). Using this combined evidence, Kuwae (Vanuatu) was used for the source and a latitude of 17°S for the latest evolv2k_v4 dataset update submitted to PANGAEA recently (Sigl & Toohey, PANGAEA, in review).

Thank you for highlighting this new work. Given that there is still sufficient uncertainty we have decided to remain conservative and have added to the paper:

*"The 1458 eruption has recently been reattributed to Kuwae in the latest eVolv2k dataset (v4) but given sufficient uncertainty we retain the unidentified classification in this work."*

Specific comments:

Figure 2: L. 301-302: Tree-rings do not reflect eruption years either, but the years of cooling which can be in the year or in the years after an eruption.

We've removed this sentence from the caption.

L. 423-24: I would argue that the tree-ring proxy network is spatially biased towards the Arctic and thus underrepresents the cooling observed in 1458 in large areas of the mid-latitudes (e.g. Central Asia, Europe, N-America). NH summer temperature reductions in 1453 from the maximum latewood density (MXD) record from Esper et al. (2017), in particular, are dominated by tree-ring sites located north of 66°N (50% of all records) in proximity to the Arctic ocean with its seasonal sea-ice cover. Temperature reductions at sites below 66°N in Esper et al., (2017) are much smaller (-1.4 °C rel. to 1961-1990) in 1453 AD and almost as large as in 1458 AD (-1.1 °C rel. to 1961-1990). This is also reflected in proxy compilations and monthly reanalysis which assimilated >170 records in the 1450s including most of the tree-ring records discussed here (Valler et al., 2024) summarized in Figures 1 and 2 (below) showing large scale cooling over Northern Hemisphere land areas in the consecutive summers of 1453/1454 and 1458/1459.

We have added the following to the discussion:

*"... it is also possible that the cooling following 1458 is underestimated in some reconstructions due to a spatial bias of individual records towards the Arctic".*

L. 592-596: there is, however, no empirical evidence that would suggest that the Samalas 1257 the 1809 or the Tambora 1815 produced strong hemispheric asymmetries of sulfate aerosols. A large network of high-quality ice-core records can be used to benchmark the spatial distribution of sulfate across both hemispheres (Sigl et al., 2014; 2015). For 1458, the spatial spread of sulfate is the opposite to your emission-based simulation, with a much larger sulfate spread over the Southern Hemisphere as could be expected for a large eruption at 17°S such as Kuwae. As it stands now it appears you are giving more credit to an apparently improved match between models and tree rings while largely ignoring existing ice-core observations suggesting otherwise.

We certainly don't want to give more credit to the model and are not suggesting that the ice-cores are incorrect. Instead, we hope to present that it could be possible for the deposition to be the same on each ice sheet but that the aerosol itself was not hemispherically symmetric – i.e. that there is not a 1:1 relationship between the stratospheric aerosol load and the average ice-sheet deposition. Given that there are numerous factors affecting the total deposition we think that this is plausible. We have added the following to the discussion:

*"Given that there are numerous factors affecting the total sulfate deposition on the ice sheets such as the large-scale and synoptic circulation and deposition processes, it is possible that the ice sheet averages could be similar, but that the stratospheric load had a greater hemispheric asymmetry"*

And the following to the conclusions when discussing future work:

*"...as well as better understanding the relationship between the stratospheric sulfate aerosol load and the amount deposited onto the ice sheets."*

L. 706-707: The most direct evidence about the spatial distribution of aerosols from past volcanic eruption arguably comes from the ice cores.

Yes, we completely agree, however, our model results still show one possible plausible reality in which the aerosol is present in both hemispheres but with a stronger asymmetry, which does not necessarily mean it is inconsistent with the ice sheet averages. Please see above for additional text added.